# Mechanical impact on neural stem cell lineage decisions in human brain organoids

Hanna Lampersperger[1,5], Michael Tranchina [1,5], Bastian Meth [1,4], Dandan Han [1], Negar Nayebzadeh[1], Nina Reiter [2], Sonja Kuth[3], Markus Lorke [3], Aldo R Boccaccini[3], Silvia Budday[2], Marisa Karow [1✉] & Sven Falk [1✉]

## Abstract

During neurodevelopment neural stem cells give rise to a spatially patterned tissue in which a regionally differentially regulated balance between proliferation and differentiation produces the fine-tuned number of neurons and macroglia necessary for a functional central nervous system. The cells driving these highly intricated developmental processes of patterning, growth and differentiation are constantly exposed to a mechanical environment that is, however, variable between different brain regions and along differentiation trajectories. Here we demonstrate that both, acute mechanical manipulations as well as a persistent change in the mechanical environment provided to human brain organoids, instruct neural stem cell lineage decisions. Furthermore, we dissect the underlying changes in the molecular program of organoid-resident cells by bulk- and single cell RNA-sequencing. These data reveal that mechanical manipulations impact on molecular programs governing early patterning events as well as cell-type-specific cellular metabolism. Thus, our results unravel a regulatory network linking mechanics and neural stem cell lineage decisions.

**Keywords** Mechanics; Neural Stem Cells; Lineage Decisions; Metabolism; Brain Organoids
**Subject Categories** Neuroscience; Stem Cells & Regenerative Medicine

See also: M Pampols-Perez & V Borrell

## Introduction

The mature central nervous system (CNS) is composed of an immense number of neurons and glial cells such as astrocytes, oligodendrocytes, and ependymal cells which all derive from a small starting population of neural stem and progenitor cells (NSCs) (Taverna et al, 2014). The orchestration of stem cell decisions that allow to expand the initial NSC pool and further allow the production of differentiated cells at the right time and place, is crucial for the accurate production of the desired cell types. The cells driving these highly intricated developmental processes of patterning, growth and differentiation are constantly exposed to a mechanical environment that is, however, variable between different brain regions (Iwashita et al, 2020) as well as changing along differentiation trajectories within the same brain region (Guo et al, 2019; Iwashita et al, 2014). The composition of the extracellular matrix (ECM) is a critical factor determining the mechanical properties of a tissue (Heisenberg and Bellaïche, 2013; LeGoff and Lecuit, 2016). It is striking that changes in the expression of ECM components, and hence the mechanical properties, accompany not only the evolutionary increase of brain size from mice to humans and the differentiation potential of NSCs within a particular species (Fietz et al, 2012; Florio et al, 2015; Long and Huttner, 2019; Long et al, 2018; Nowakowski et al, 2017; Pollen et al, 2015), but also vary dramatically between different brain regions (Amin and Borrell, 2020; Copp et al, 2011; Dauth et al, 2016).

Correlating with the change in the mechanical properties along neuronal differentiation trajectories also the metabolic profiles of the differentiating cells are shifting (Knobloch and Jessberger, 2017) linking mechanics with the metabolism of a cell through guilt-by-association. Furthermore, evidence that mechanics, metabolism and regional identity are likely interwoven, is provided by recent findings showing that the cellular energy metabolism is spatially patterned during early morphogenetic events (reviewed in: Lemma and Nelson, 2023), events that are controlled and strongly influenced by mechanical forces (Abdel Fattah et al, 2021; Heisenberg and Bellaïche, 2013; Vicente and Diz-Muñoz, 2023). Strikingly, the acquisition of regional identity during patterning processes of the early human brain impacts on the cellular metabolism (Fleck et al, 2023), in particular lipid metabolism, putting the varying mechanical properties found in different brain regions into context with local differences in the cellular metabolism. Interestingly, cellular mechanics and cellular metabolism are feedback-regulated processes (Evers et al, 2021) each dependent on the other.

With the advent of directing human induced pluripotent stem cells to three-dimensional brain organoids recapitulating key features of early human brain development a new era dawned (Velasco et al, 2020). These brain organoids provide a scalable,

[1]Institute of Biochemistry, Friedrich-Alexander-Universität Erlangen-Nürnberg, Erlangen, Germany. [2]Institute of Continuum Mechanics and Biomechanics, Friedrich-Alexander-Universität Erlangen-Nürnberg, Erlangen, Germany. [3]Institute of Biomaterials, Friedrich-Alexander-Universität Erlangen-Nürnberg, Erlangen, Germany. [4]Present address: Department of Functional Genomics, Center for Neurogenomics and Cognitive Research, Vrije Universiteit Amsterdam, Amsterdam, The Netherlands. [5]These authors contributed equally: Hanna Lampersperger, Michael Tranchina. ✉E-mail: marisa.karow@fau.de; sven.falk@fau.de

easily accessible, and tunable system of human neurodevelopment allowing for a fine-grained control of the environment the developing brain tissue is exposed to. Here, we set out to uncouple the biochemical and biophysical properties of the cellular environment to assess the consequences of applying either an acute mechanical impact on brain organoids or providing a persistent change in the mechanical environment of organoids. We hereby assessed both the consequences on a cellular as well as a molecular level and discovered a hitherto unknown level of regulation of early neurodevelopment. We uncovered that mechanical stimuli influence not only the balance between proliferation and differentiation of NSCs but also affect molecular programs directing regional identity and metabolism of neural cells.

## Results and discussion

### Acute compression of brain organoids results in SOX2 upregulation

To set up a system allowing to assess the impact of mechanical forces on NSCs in a human developmental setting, we generated brain organoids from human induced pluripotent stem cells (hiPSC) (Lancaster et al, 2013). We used brain organoids at d30 following aggregation of the hiPSCs to perform unconfined large-strain (Reiter et al, 2021) mediated compression experiments using a rheometer (Fig. 1A). At d30 of organoid development ventricular zone-like structures (VZLS) containing neural stem and progenitor cells (SOX2 positive) as well as neuronal compartments (NC) composed of postmitotic neurons (MAP2 positive) are clearly discernable (Fig. EV1A). The rheometer allows not only to compress a sample precisely by prescribing the desired deformation but also to directly measure the force acting on the brain organoids during compression. For mechanical manipulations, organoids were placed on the rheometer platform and compressed in three consecutive cycles (by 40%, 50%, 60% of its initial height) with each compression cycle lasting roughly 1 min. In addition to the control (ctrl) group, i.e., organoids that were cultured and not manipulated we included mock manipulated organoids (mock), i.e., organoids that were placed on the rheometer platform the same way as the compressed organoids but were not compressed. We combined both groups together as 'uncompressed'. For each compressed organoid, we determined the acting force for the entire duration of each compression cycle (Fig. EV1B). During the second and third cycle of compression, the forces acting on the organoid are typically lower. After compression, we kept the organoids separately and incubated them for another 24 h (h) or up to 5 days (d). Using immunofluorescence (IF) stainings of organoid slices, we performed tissue phenotyping with a first focus on organoid resident progenitors expressing the neural stem and progenitor marker SOX2. In addition, we used Phalloidin to visualize F-actin which allows to assess the integrity of the VZLS particularly at the apical side. While we did not observe morphological alterations at the apical site of the VZLS, we found a prominent increase in the overall presence of SOX2 signal across the entire sections of compressed organoids (Fig. 1B). To quantify this effect, we established a semi-automated image analysis pipeline that allows for compartmentalization into VZLS and NC areas (Fig. EV1C). Employing this pipeline we detected a compression-induced

gradual increase in the SOX2 protein levels in whole organoid sections (Fig. 1C), raising the question whether this is directly correlated with the mechanical forces acting on the respective organoids. We therefore computed the linear regression between the SOX2 protein level in each tested organoid and the average maximum peak force acting on it (Fig. 1D). This analysis shows a significant correlation between the average maximum peak force and the levels of SOX2. Furthermore, sequential testing of the correlation between the peak forces of each of the three consecutive compression cycles amongst each other, showed that there is no change in the correlation (Fig. EV1D), i.e., forces in the first compression cycle strongly correlated with the ones in the second cycle and the ones in the third cycle and so did the forces of the second and the third cycle. These correlations indicate that viscoelastic effects leading to the reduced peak force in cycle 2 and 3 (Fig. EV1B) are consistent across organoids.

Next, we further dissected the localization of the SOX2 upregulating cells. First, we subsetted the NSC containing VZLS and found a similar increase of SOX2 protein levels within VZLS (Fig. 1E). We then binned the apico–basal axis of the VZLS in ten equally sized bins with bin 0 at the most apical and bin 9 at the most basal site and compared the expression levels in uncompressed and compressed samples. These analyses revealed that across the apico–basal axis SOX2 protein levels are induced to a similar extent independent of the relative localization of the cells (Fig. 1F). To test for how long after compression SOX2 protein levels are changed, we assessed brain organoids 5 d post compression. At this stage, no significant changes between the experimental conditions could be observed, neither in the whole organoid sections (Fig. 1G) nor in the VZLS only (Fig. EV1E), showing that upregulation of SOX2 is transient.

In the here described large-strain compression experiments, we compress brain organoids with peak forces between 2 mM and 14 mN (Fig. EV1D). Assuming an organoid diameter of 3 mm, this translates to nominal stress values between 282 Pa and 1980 Pa. During in utero natural brain development of mice the intracranial pressure oscillates within a range of 150–1500 Pa with an oscillation period of around 33 s (Akaike et al, 2025) indicating dynamic mechanical stimuli to an extent comparable to the duration and magnitude of manipulations performed in our study. The intracranial pressure is driven by uterine contractions as well as the production of the cerebrospinal fluid (CSF) (Jones et al, 1987; Moazen et al, 2016). In chick embryos release of CSF, and by that both changing the pressure as well as the biochemical milieu, alters proliferation of progenitors in the developing CNS (Desmond and Schoenwolf, 1985; Gato et al, 2005). Using rheometer-mediated compression, we now can disentangle the impact of mechanical aspects from biochemical factors impacting on cellular behavior. The forces applied during rheometer-mediated compressions (Fig. EV1B) are in the same magnitude as the ones described in vivo. Hence, the rheometer-mediated compression allows to apply forces similar to the ones acting during brain development but for a limited time and temporally precisely controlled.

### Acute compression of older brain organoids does not alter SOX2 levels

Next, we wondered about the impact of an acute compression of older brain organoids, in which the overall existence of SOX2

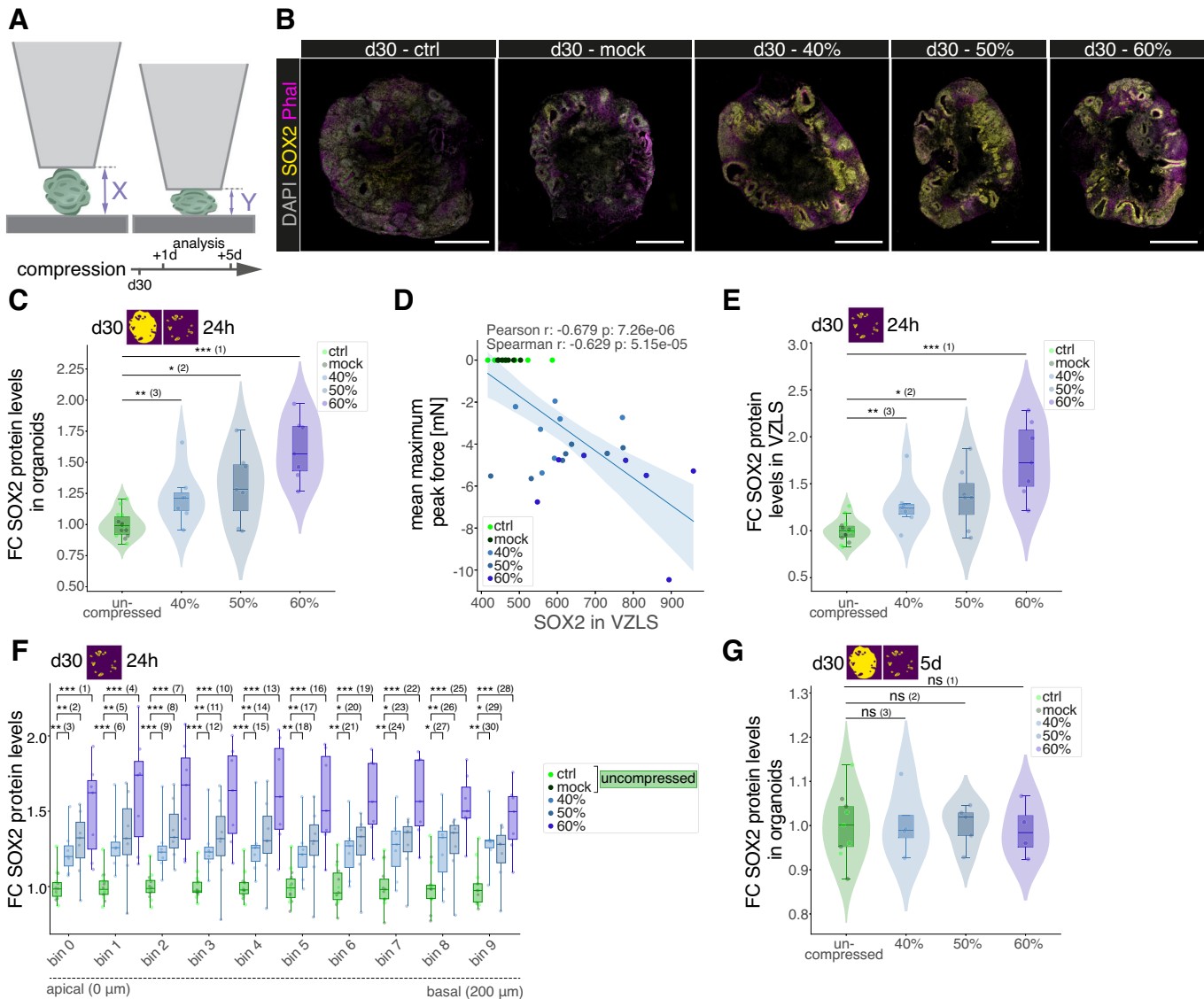

**Figure 1. Acute compression of brain organoids results in SOX2 upregulation.**

(A) Schematic depiction of rheometer-mediated compression and measurement of forces on brain organoids including the timeline of experiments. (B) Images showing immunofluorescence (IF) stainings of slices from d30 brain organoid 24 h after compression across experimental conditions. Phalloidin (magenta) was used to visualize F-actin, prominent at the apical side within VZLS. Note the gradually increasing intensity of the SOX2 signal (yellow) in compressed organoids. DAPI was used to counterstain nuclei. Scale bar = 500 μm. (C) Violin, box, and jitter plots showing the fold change (FC) in mean SOX2 signal in SOX2-positive cells in whole organoid sections 24 h after compression, relative to uncompressed organoids, based on IF intensity measurements. Ctrl: $n = 7$; mock: $n = 7$, 40%: $n = 7$; 50%: $n = 7$; 60%: $n = 7$. Exact P values: (1): $1.720 \times 10^{-5}$; (2): $2.000 \times 10^{-2}$; (3): $4.231 \times 10^{-3}$. (D) Linear regression showing the correlation between the mean maximum peak force (mean of the maximum peak forces in the three compression cycles) acting on individual organoids and the mean SOX2 protein expression in SOX2-positive cells in the VZLS 24 h after compression, based on IF intensity measurements. Ctrl: $n = 7$; mock: $n = 7$, 40%: $n = 7$; 50%: $n = 7$; 60%: $n = 7$. (E) Violin, box, and jitter plots showing the fold change (FC) in mean SOX2 signal in SOX2-positive cells in the VZLS 24 h after compression, relative to uncompressed organoids, based on IF intensity measurements. Ctrl: $n = 7$; mock: $n = 7$, 40%: n = 7; 50%: $n = 7$; 60%: $n = 7$. Exact P values: (1): $3.440 \times 10^{-5}$; (2): $2.000 \times 10^{-2}$; (3): $7.448 \times 10^{-3}$. (F) Box and jitter plots showing SOX2 protein levels across conditions in equally sized 20 μm bins along the apico–basal axis of the VZLS, depicted as fold change (FC) relative to the mean SOX2 protein levels in uncompressed samples within the same bin, based on IF intensity measurements. Ctrl: $n = 7$; mock: $n = 7$, 40%: $n = 7$; 50%: $n = 7$; 60%: $n = 7$. Exact P values: (1): $6.9 \times 10^{-5}$; (2): $7.4 \times 10^{-3}$; (3): $1.6 \times 10^{-3}$; (4): $6.9 \times 10^{-5}$; (5): $9.7 \times 10^{-3}$; (6): $2.1 \times 10^{-4}$; (7): $6.9 \times 10^{-5}$; (8): $3.4 \times 10^{-5}$; (9): $5.2 \times 10^{-4}$; (10): $3.4 \times 10^{-5}$; (11): $9.7 \times 10^{-3}$; (12): $3.3 \times 10^{-4}$; (13): $3.4 \times 10^{-5}$; (14): $9.7 \times 10^{-3}$; (15): $2.1 \times 10^{-4}$; (16): $6.9 \times 10^{-5}$ (17): $9.7 \times 10^{-3}$; (18) $7.4 \times 10^{-3}$; (19): $3.4 \times 10^{-5}$; (20): $1.2 \times 10^{-2}$; (21) $4.2 \times 10^{-3}$; (22): $6.9 \times 10^{-5}$; (23): $1.2 \times 10^{-2}$; (24): $5.6 \times 10^{-3}$; (25): $3.4 \times 10^{-5}$; (26): $9.7 \times 10^{-3}$; (27): $4.6 \times 10^{-2}$; (28): $1.8 \times 10^{-4}$; (29): $2.9 \times 10^{-2}$; (30): $6.6 \times 10^{-3}$. (G) Violin, box, and jitter plots showing the fold change (FC) in mean SOX2 signal in SOX2-positive cells in whole organoid sections 5 days after compression, relative to uncompressed organoids, based on IF intensity measurements. Ctrl: $n = 5$; mock: $n = 4$, 40%: $n = 4$; 50%: $n = 5$; 60%: $n = 4$; exact P values: (1): 1.0; (2): 1.0; (3): 0.939. For (C, E, G), violin plots show the distribution of the data. For (C, E, F, G), boxplots show the median (center line), interquartile range (box), and whiskers extending to the most extreme values within 1.5× the interquartile range; dots represent individual organoids. Statistical significance was assessed using a two-sided Wilcoxon rank-sum test (ns: $P > 0.05$; *: $P < 0.05$; **: $P < 0.01$; ***: $P < 0.001$). Yellow/magenta images in the upper left corner indicate whether VZLS and/or NC were included (according to the segmentation shown in Fig. EV1C). Source data are available online for this figure.

expressing NSCs is lower and more mature neurons are present (Quadrato et al, 2017). We therefore performed the same compression experiments in 60- and 70-day-old organoids (Fig. 2A). We here found that in contrast to d30 organoids, 50% compression did not result in a change of SOX2 levels compared to uncompressed samples (Fig. 2B,C). Also, when we dissected the SOX2 protein levels in VZLS (Fig. 2D) and NCs (Fig. 2E), no significant changes in the compressed samples at both timepoints were discerned. We further found no change in the fraction of SOX2-expressing cells upon compression (Fig. 2F). These data suggest that there is a temporal window of susceptibility in which the protein levels of the NSC controlling factor SOX2 are modulated by mechanical forces. Importantly, we performed the same compression with the same degree of deformation at all compared developmental stages. Absence of cellular response to compression can hence not be explained by varying degrees of deformation.

## Persistent changes in the physical environment as well as acute mechanical impact increase the fraction of SOX2 and impact on neuronal differentiation

We observed a transient upregulation of the NSC factor SOX2 after compression with more pronounced effects at the apical site of the VZLS. At the apical site, a NSC either divides symmetrically proliferative to produce two new NSCs, asymmetrically neurogenic to produce a NSC and a neuron, and symmetrically neurogenic to produce two neurons (Taverna et al, 2014). We therefore asked whether mechanical manipulation of NSCs not only influences the expression of the NSC marker SOX2 but also impacts on the lineage decisions of NSCs. We first quantified the fraction of organoid area occupied by SOX2+ NSCs and found that 24 h after compression, this fraction increased in a compression-dependent manner (Fig. 3A). Given that at this stage of development NSCs either produce NSCs or neurons, we next assessed whether this increase in SOX2-positive areas occurs at the expense of neuron production. To this end we added the thymidine analog BrdU directly after compression to label cells which went through S-phase after compression and follow their offspring. We analyzed the organoids 5 days later (Fig. 3B) by co-staining for the neuronal marker NEUN together with SOX2 and BrdU (Fig. EV2A), thereby quantifying the fraction of cells which underwent differentiation into postmitotic neurons after compression. Interestingly, we found a significant decrease in the fraction of NEUN +/BrdU+ cells over total BrdU+ cells (Fig. 3C), showing that mechanical manipulation of organoids not only results in a transient upregulation of SOX2, but also reduces the output of NEUN+ neurons from neural stem and progenitor cells. We next asked whether large-strain compression of organoids induced cell death. To this end we stained sections of compressed organoids treated for 5 days with BrdU for the apoptotic marker cleaved-Caspase 3 (cCAS3), BrdU, and NEUN. These analyses show that there is neither a significant difference in apoptosis in all cells born after compression (cCAS3/BrdU; Fig. EV2B,C) nor in young neurons born after compression (cCAS3/BrdU/NEUN; Figs. 3D and EV2B). Hence, a cell type-specific increase in cell death can be ruled out as the underlying explanation for the changes in the cellular composition after mechanical manipulation. Together, these data highlight a so far unappreciated level of NSC regulation where

mechanical stimuli influence the balance between proliferation and differentiation of NSCs.

Motivated by the observation that acute mechanical manipulations change SOX2 protein levels and influence NSC lineage decisions, we set out to examine the effect of a persistent change in the mechanical environment on early human brain development. During the canonical organoid protocol, the organoids are typically embedded in Matrigel on d6-8 following neural induction (Lancaster et al, 2013). To change the mechanical properties, one would have to change Matrigel concentrations which will not only affect the mechanical environment but also alter the composition of the biochemical environment (e.g., concentration of growth factors), rendering it difficult to disentangle the contribution of the mechanical properties on cellular phenotypes. Moreover, Matrigel degrades over time with no discernable Matrigel being left around organoids at d30. To overcome these limitations, we utilized hydrogels (HG) built from oxidized hyaluronic acid (OHA) and gelatin crosslinked by microbial transglutaminase (Fig. 3E,F) (Kuth et al, 2022). These HGs can be reproducibly fabricated and finetuned to a specific stiffness in contrast to the batch-to-batch varying 200–600 Pa estimated for Matrigel (Aisenbrey and Murphy, 2020). We employed 1.25%OHA/2.5%GEL (referred to as HG1) with 230 Pa and 2.5%OHA/2.5%GEL (referred to as HG2) with 500 Pa initial stiffness, resulting in proper organoid formation in all conditions (Fig. 3G). We found a substantial increase in the fraction of SOX2 per organoid, dependent on the stiffness of the environment the neural tissue was developing in (Fig. 3G,H). Yet, the SOX2 protein levels remained unchanged within the VZLS (Fig. 3I) as well as in the NC (Fig. 3J). Generally, the variations in SOX2 protein levels are higher in the HG than in the MG embedded organoids. The hydrogels are stiffer than the Matrigel typically used for embedding. Presumably, the stiffer the matrix on the outside, the steeper the gradient inside the organoid. This likely leads to a non-uniform response of the cells to the stiffer surrounding matrix, i.e., the closer the cells are to the outer boarder of the organoid, the more pronounced the response. If this would be the case, then this could lead to a bigger variation across samples. From these results we conclude that the stiffness of the environment controls the number of SOX2-expressing NSCs but unlike after acute mechanical manipulations, does not alter the average protein levels of SOX2 in organoids.

In sum, our data provide evidence that acute mechanical manipulations impact on neural stem cells in developing brain organoids likely mediated through changes in the levels of the NSC factor SOX2. These results prompted us to investigate the molecular changes induced by the mechanical impact early upon compression and to find potential molecular reasons for the increase in NSCs and the decrease of neurons.

## Global transcriptome changes in compressed organoids

To uncover the global transcriptomic changes induced by compression we performed bulk RNA-sequencing (bulk RNA-seq) of uncompressed organoids and organoids compressed to either 50% or 60% after 24 h. Principal component analysis revealed a striking separation of samples by the experimental condition in PC1, where the highest difference in molecular profiles across samples can be found (Fig. 4A). We next computed the differentially expressed genes between uncompressed, 50%

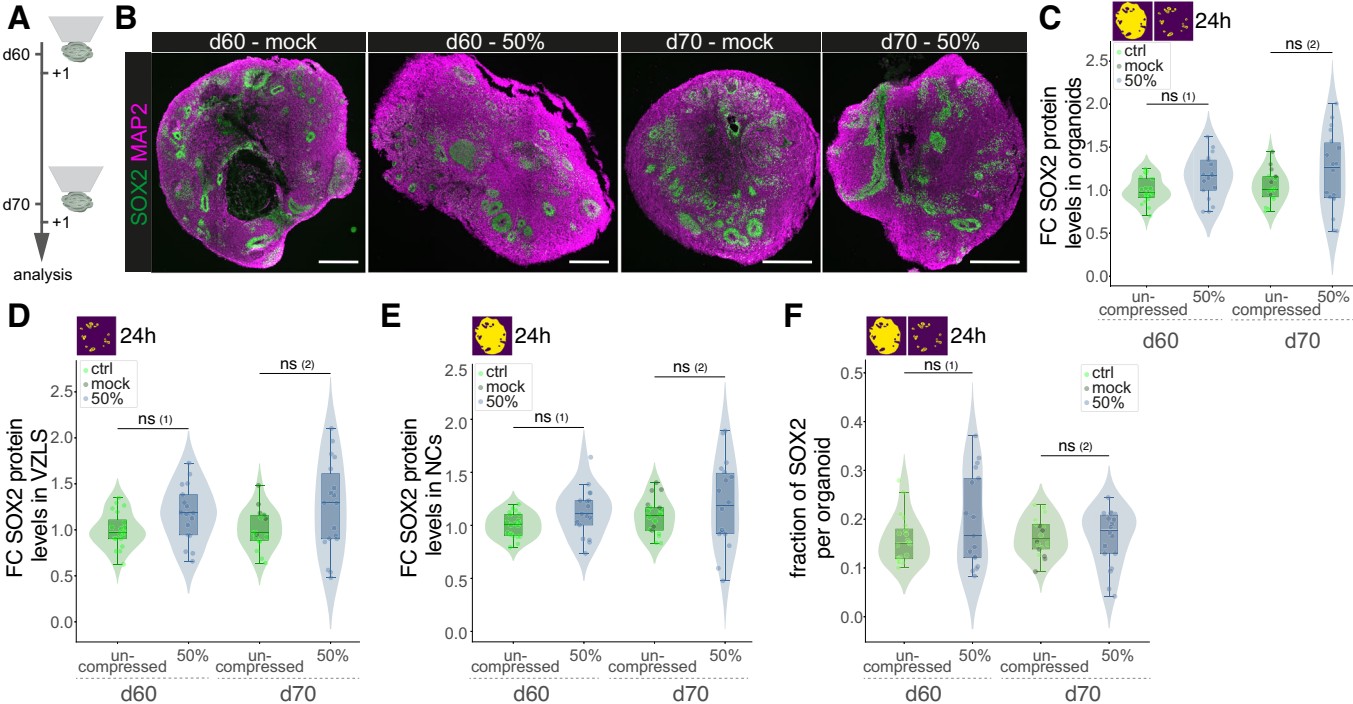

**Figure 2. Acute compression of older brain organoids does not alter SOX2 levels.**

(A) Experimental outline of rheometer-mediated compression and measurement of forces on older brain organoids including the timeline of experiments. (B) Pictures showing immunofluorescence (IF) stainings of SOX2 (green) and MAP2 (magenta) in 60 d and 70 d mock and 50% compressed brain organoid samples. Scale bar = 500 μm. (C) Violin, box, and jitter plots showing the fold change (FC) in mean SOX2 signal in SOX2-positive cells in whole organoid sections 24 h after compression of d60 and d70 old organoids, relative to uncompressed organoids of the same age, based on IF intensity measurements. Exact P values: (1): $6.521 \times 10^{-1}$; (2): $2.614 \times 10^{-1}$. (D) Violin, box, and jitter plots showing the fold change (FC) in mean SOX2 signal in SOX2-positive cells in the VZLS 24 h after compression of d60 and d70 old organoids, relative to uncompressed organoids of the same age, based on IF intensity measurements. Exact P values: (1): $7.461 \times 10^{-1}$; (2): $1.687 \times 10^{-1}$. (E) Violin, box, and jitter plots showing the fold change (FC) in mean SOX2 signal in SOX2-positive cells in the NC 24 h after compression of d60 and d70 old organoids, relative to uncompressed organoids of the same age, based on IF intensity measurements. Exact P values: (1): $6.089 \times 10^{-2}$; (2): $4.383 \times 10^{-1}$. (F) Violin, box, and jitter plots showing the fraction of SOX2 positive areas per whole organoid sections in d60 and d70 old organoids 24 h after compression. Exact P values: (1): $4.371 \times 10^{-1}$; (2): $8.868 \times 10^{-1}$. For (C–F), ctrl d60: $n = 20$; 50% comp. d60: $n = 17$; ctrl d70: $n = 11$; mock d70: $n = 7$; 50% comp. d70: $n = 18$. Violin plots show the distribution of the data. Boxplots show the median (center line), interquartile range (box), and whiskers extending to the most extreme values within 1.5× the interquartile range; dots represent individual organoids. Statistical significance was assessed using a two-sided Wilcoxon rank-sum test (ns: $P > 0.05$; *: $P < 0.05$; **: $P < 0.01$; ***: $P < 0.001$). Yellow/magenta images in the upper left corner indicate whether VZLS and/or NC were included (according to the segmentation shown in Fig. EV1C). Source data are available online for this figure.

compression, and 60% compression samples and checked how many of these genes are shared between different conditions. As evident in the upset plot (Fig. 4B), we found that the vast majority of genes differentially expressed due to compression are shared between 50% and 60% compression samples (1425) and very few genes are specifically deregulated due to the increase of compression from 50% to 60% (48) (Fig. 4B). This analysis shows that the molecular changes induced by mechanical manipulations largely affect the same transcriptional programs independent of the extent of the compression. To unbiasedly reveal the cellular processes that are affected by compression we performed a gene ontology (GO) enrichment analysis based on the differentially expressed (DE) genes between the mock and the compressed samples. Interestingly, GO terms associated with lipid metabolism ('very-low-density lipoprotein particle clearance', 'tryglyceride catabolic processes', or 'cholesterol biosynthetic process') were enriched as well as GO terms indicating patterning such as 'forebrain regionalization' (Fig. 4C). To further elucidate how brain regionalization is affected by mechanical manipulations, we plotted

individual genes differentially expressed and implicated in dorso-ventral as well as anterior-posterior patterning processes. This analysis revealed an upregulation of ventral genes such as *GSX2, WNT10A* along with downregulation of dorsal genes such as *TBR1, BMP4,* and *TTR* (Kelly et al, 1993; Pellegrini et al, 2020; Renner et al, 2017) (Fig. 4D). Similarly, genes associated with anterior-posterior patterning such as *HOX* genes (Saito and Suzuki, 2020) or *GBX1/GBX2* (Wassarman et al, 1997) were deregulated (Fig. 4D). These analyses support the notion that mechanical manipulation results in the induction of more ventral and more posterior transcriptional programs. Together these findings prompted us to determine the transcriptional deviation in signaling pathways implicated in the establishment of main axes of the developing CNS as well as pathways involved in controlling the metabolism of cells employing the KEGG pathway annotations (Kanehisa, 2000; Kanehisa, 2019; Kanehisa et al, 2023). While pathways associated with more general terms such as 'basal transcription' or 'RNA polymerase' were not changed in any of the conditions as expected (Fig. 4E), we found strong deregulation of pathways instructing

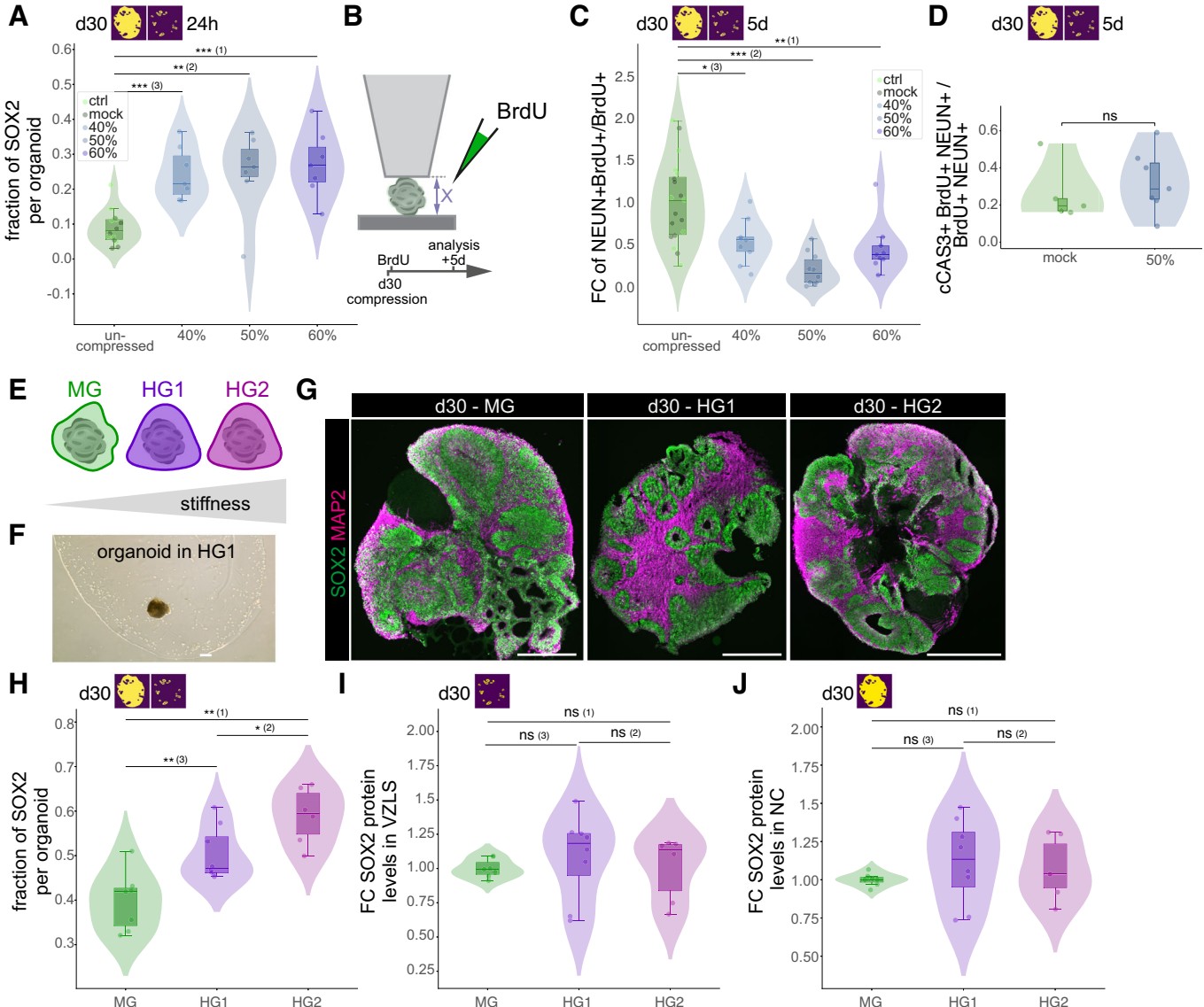

Figure 3. Persistent changes in the physical environment as well as acute mechanical impact increase the fraction of SOX2 and impact on neuronal differentiation.

(A) Violin, box, and jitter plots showing the fraction of SOX2-positive areas per whole organoid sections 24 h after compression of d30 organoids. Ctrl: n = 7; mock: n = 7, 40%: n = 7; 50%: n = 7; 60%: n = 7. Exact P values: (1): 1.204 × 10⁻⁴; (2): 7.448 × 10⁻³; (3): 1.204 × 10⁻⁴. (B) Experimental scheme depicting the BrdU lineage tracing experiments to reveal cells born after the compression. (C) Violin, box, and jitter plots showing the fold change (FC) relative to uncompressed organoids in the fraction of BrdU/NEUN double positive cells among total BrdU positive cells, i.e., cells that went through S-phase after compression (BrdU +) and left cell cycle to differentiate into neurons (NEUN +) within 5 days after compression. Ctrl: n = 9; mock: n = 10, 40%: n = 9; 50%: n = 10; 60%: n = 9. Exact P values: (1): 5.334e-03; (2): 3.598e-05; (3): 1.308e-02. (D) Violin, box, and jitter plots showing the fraction of cleaved caspase 3 (cCAS3)/BrdU/NEUN-triple positive cells among all BrdU/NEUN double positive cells to reveal the fraction of apoptotic cells among the neurons born after compression. Mock: n = 5, 50%: n = 7. Exact P value: 0.34. (E) Experimental scheme illustrating differences in stiffness between Matrigel (MG) and hydrogels HG1 and HG2. (F) Brightfield picture showing a brain organoid embedded in HG1. Scale bar = 500 μm. (G) Immunofluorescence (IF) images showing SOX2 positive progenitors (green) and MAP2 positive neurons (magenta) in sections of d30 brain organoids embedded in MG (left) or hydrogels HG1 and HG2 (middle and right). Scale bar = 500 μm. (H) Violin, box, and jitter plots showing the fraction of SOX2 positive areas per whole organoid sections of d30 organoids embedded in MG or hydrogels HG1 and HG2. Exact P values: (1): 2.331 × 10⁻³; (2): 2.930 × 10⁻²; (3): 5.905 × 10⁻³. (I) Violin, box, and jitter plots showing the fold change (FC) in mean SOX2 signal in SOX2-positive cells in the VZLS of d30 organoids embedded in different matrixes, relative to MG embedded organoids, based on IF intensity measurements. Exact P values: (1): 3.660 × 10⁻¹; (2): 5.728 × 10⁻¹; (3): 1.893 × 10⁻¹. (J) Violin, box, and jitter plots showing the fold change (FC) in mean SOX2 signal in SOX2-positive cells in the NC of d30 organoids embedded in different matrixes, relative to MG embedded organoids, based on IF intensity measurements. Exact P values: (1): 5.338 × 10⁻¹; (2): 8.518 × 10⁻¹; (3): 2.319 × 10⁻¹. For (G, H, I, J), MG = matrigel; HG1 = 1.25% OHA/2.5% GEL; HG2 = 2.5% OHA/2.5% GEL; MG: n = 6; HG1: n = 8; HG2: n = 6. For (A, C, D, H–J), Violin plots show the distribution of the data. Boxplots show the median (center line), interquartile range (box), and whiskers extending to the most extreme values within 1.5× the interquartile range; dots represent individual organoids. Statistical significance was assessed using a two-sided Wilcoxon rank-sum test (ns: P > 0.05; *: P < 0.05; **: P < 0.01; ***: P < 0.001). Yellow/magenta images in the upper left corner indicate whether VZLS and/or NC were included (according to the segmentation shown in Fig. EV1C). Source data are available online for this figure.

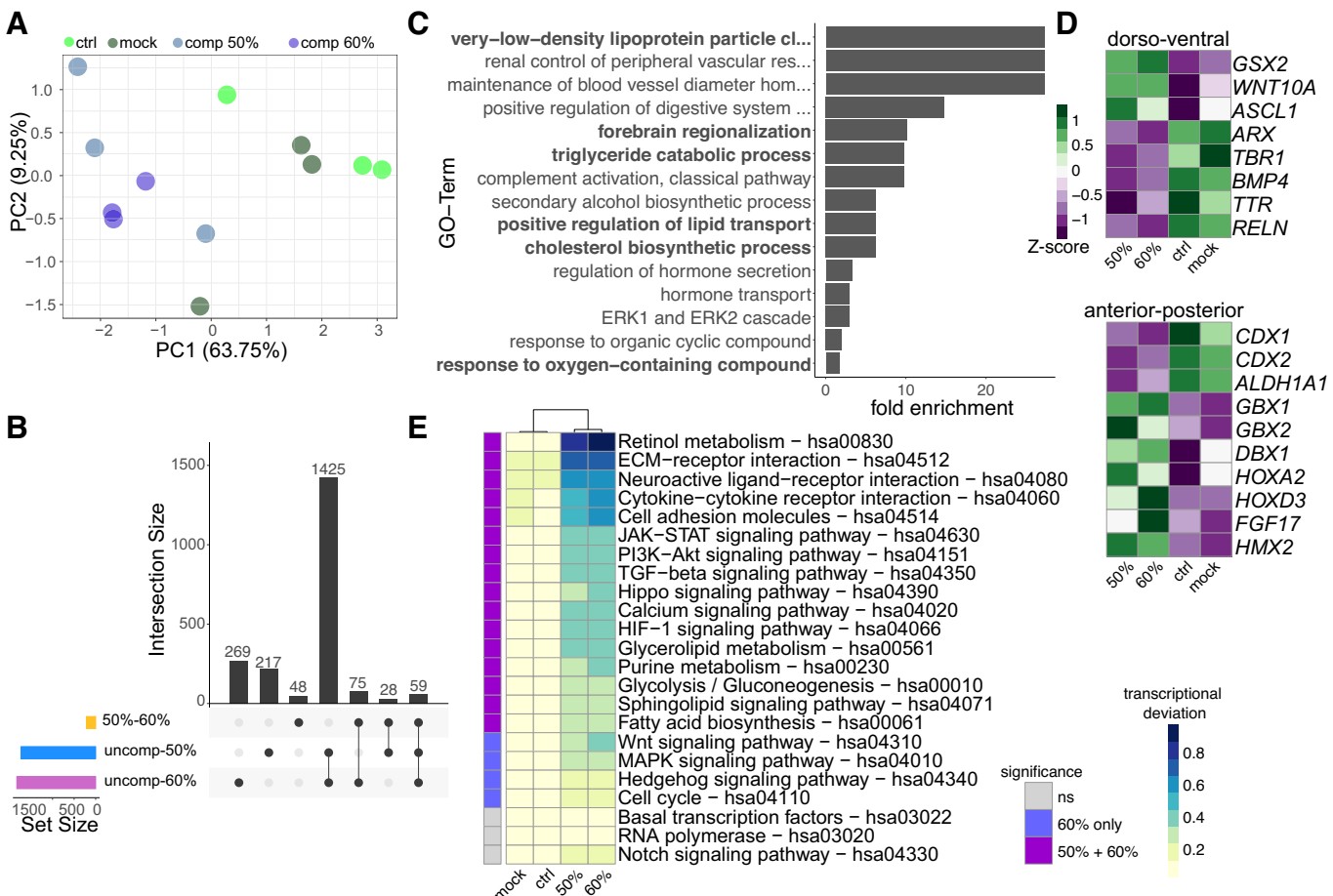

**Figure 4. Global transcriptome changes in compressed organoids.**

(A) Principal component analysis of bulk RNA-seq samples based on all expressed genes. Experimental IDs are indicated by the color. (B) Upset plot showing the intersection of differentially expressed (DE) genes between the differently compressed samples (50–60%) and the compressed samples individually with the uncompressed samples (uncomp−50% and uncomp−60%). Numbers of genes in each intersection are indicated above the bars. (C) Top 15 GO terms of differentially expressed genes between mock and 50% compressed samples. (D) Heatmap showing deregulation of patterning-associated genes indicating changes in dorso-ventral (upper panel) and anterior-posterior (lower panel) patterning in compressed samples. Z-scores are shown. (E) Heatmap showing the transcriptional deviation of genes within selected KEGG pathways across samples. The left column indicates whether the deviation in the respective category is significant ($p < 0.05$; hues of purple) and whether it is significant either only in 60% compressed samples or both compressed conditions (50% and 60%) or not significantly changed ($p > 0.05$; gray). Significance was tested using the one-way ANOVA with Tukey's post hoc test between uncompressed samples and the respective compression sample. Dendrogram shows similarity between samples.

patterning process such as retinol metabolism, TGF-beta signaling, WNT signaling, and hedgehog signaling pathways in conjunction with alterations in pathways controlling cellular metabolism (Fig. 4E). Of note, while there were minor quantitative differences between 50% and 60% compressed samples, only few of the tested KEGG pathways showed significant deregulation unique to the 60% compression conditions (Fig. 4E), reiterating the finding that molecular differences between 50% and 60% compressed samples is low. Regulation of the activity of the WNT pathway by different levels of SOX2 levels is in line with earlier findings showing that depending on the amount of SOX2, distinct WNT responses triggering differentiation in pluripotent stem cells are elicited (Blassberg et al, 2022).

In sum, the bulk transcriptome data unraveled, that acute mechanical manipulations impact on gene regulatory networks governing cellular metabolism and patterning processes.

## Molecular dissection of the impact of compression on a single cell level

Based on our findings that acute mechanical insults can induce transcriptional programs expressed in specific brain regions, we set out to deconstruct the molecular changes on a single cell level by using single-cell RNA-sequencing (scRNA-seq). We took advantage of the heterogenous nature of cell types and lineages within unguided brain organoids. To avoid the observation of line-specific effects, we multiplexed compressed organoids generated by two different hiPSC lines (XM001RFP and wibj2 as detailed in the method section). Brain organoids were compressed (50%) on d30 post aggregation of the hiPSCs and analyzed 24 h later. We used two mock samples as controls each presenting a pool of 3 organoids (1 XM001RFP pool, 1 wibj2 pool), and 4 compressed organoids (2 XM001RFP, 2 wibj2). As evident in the force-directed graph embedding (Figs. 5A and EV3A,B), we

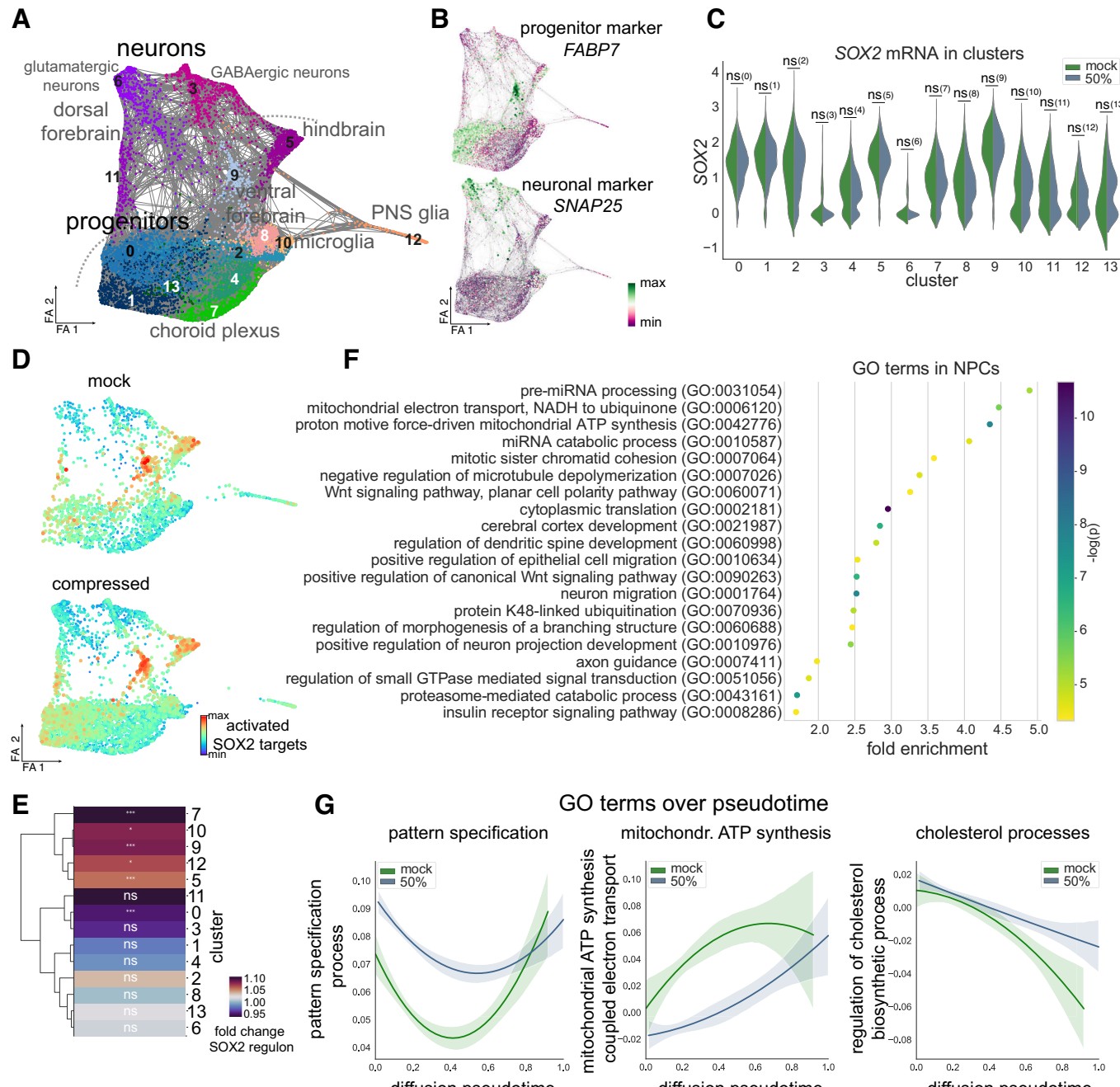

detected neural stem and progenitor cells (*FABP7*, Fig. 5B; *SOX2*, *NES*, *HES5*, *TOP2A*, Fig. EV3D) and neurons (*SNAP25*, Fig. 5B; *DCX*, Fig. EV3C) stratified in distinct lineages giving rise to PNS glia (Fig. EV3E), dorsal forebrain (Fig. EV3F), ventral forebrain (Fig. EV3G,H), hindbrain (Fig. EV3I), microglia (Fig. EV3J), and choroid plexus (Fig. EV3K). Both hiPSC lines used contribute similarly to the different clusters and trajectories (Fig. EV3B). Importantly, compression of organoids did not result in novel clusters, or a significant alteration in the distribution to clusters as computed by scCODA (Büttner et al, 2021), upon compression but rather resulted in more subtle transcriptional changes with compressed cells clustering together with mock cells (Fig. EV3A).

We next scrutinized the scRNA-seq data to assess whether also on mRNA level the *SOX2* expression was increased upon compression. However, in none of the clusters we detected a significant increase of SOX2 mRNA in compressed cells compared to mock cell (Fig. 5C) revealing that mechanical manipulation regulates SOX2 protein levels on a posttranscriptional level. When we then focused on the transcriptional network controlled by the transcription factor SOX2 we found a profound difference in the expression of genes activated by SOX2 across conditions (Fig. 5D) in line with the observation that SOX2 protein levels are increased in compressed organoids (Fig. 1). However, the impact on the SOX2 regulatory network by mechanical manipulation is cell-type

◄

**Figure 5. Molecular dissection of the impact of compression on single-cell level.**

(A) Force-directed graph embedding showing a total of 7.791 cells (mock: 2.344 cells; 50% compressed: 5.447 cells) derived from organoids of 2 independent human iPSC lines and analyzed by scRNA-seq. Distinct lineages are highlighted and corresponding plots depicting specific lineage markers can be found in Fig. EV3. Leiden cluster numbers are indicated on the plot. (B) Expression of the progenitor marker *FABP7* (upper plot) and the neuronal marker *SNAP25* (lower plot) projected onto the embedding shown in (A). (C) Violin plot showing the SOX2 mRNA expression in each cluster in mock and 50% compressed cells. Of note, no significant difference in SOX2 mRNA could be detected in any cluster. Exact $P$ values: (0): 0.242; (1): 0.923; (2): 0.510; (3): 0.166; (4): 0.443; (5): 0.973; (6): 0.299; (7): 0.934; (8): 0.068; (9): 0.610; (10): 0.088; (11): 0.116; (12): 0.460; (13): 0.381. Number of cells in each cluster (mock/50% compressed): (0): 220/1175; (1): 324/819; (2): 99/561; (3): 229/390; (4): 128/491; (5): 295/312; (6): 125/427; (7): 262/289; (8): 213/331; (9): 168/220; (10): 126/105; (11): 53/173; (12): 97/57; (13): 5/97. (D) Expression score of genes activated by SOX2 projected onto embedding shown in A and separated by condition. Upper panel showing mock cells, lower panel showing 50% compressed cells. (E) Heatmap indicating the extent of the change in the SOX2 regulon activity in compressed versus uncompressed cells in the different clusters show in plot A. Dendrogram on the left illustrates similarity across the clusters. Exact $P$ values: (7): $4.4 \times 10^{-8}$; (10): $1.1 \times 10^{-2}$; (9): $2.4 \times 10^{-4}$; (12): $2.5 \times 10^{-2}$; (5): $1.3 \times 10^{-4}$; (11): $9.0 \times 10^{-2}$; (0): $9.6 \times 10^{-6}$; (3): $1.5 \times 10^{-1}$; (1): $1.5 \times 10^{-1}$; (4): $3.5 \times 10^{-1}$; (2): $9.8 \times 10^{-2}$; (8): $8.0 \times 10^{-1}$; (13): $8.1 \times 10^{-1}$; (6):$8.8 \times 10^{-1}$. (F) GO terms associated with differentially expressed genes between mock and compressed progenitor cells (cluster 0, 1, 5, 9). (G) Fitted line plot indicating the average expression (score) of the gene set associated with the respective GO terms (Y-axis) along diffusion pseudotime (X-axis) in mock and 50% compressed cells. 95% confidence interval is shown as a transparent band in the respective same color. All neural stem and progenitor cells as well as the neurons derived from them are included (cluster 0, 1, 3, 5, 6, 9, 11, 13 in Fig. 5A). For (A, B, D), FA refers to force atlas. For (C, E), Two-sided Wilcoxon rank-sum test (ns: $P > 0.05$; *: $P < 0.05$; **: $P < 0.01$; ***: $P < 0.001$).

specific showing brain-regional differences. As evident from the heatmap (Fig. 5E), particularly clusters with ventral forebrain progenitor (cluster 9), hindbrain progenitor (cluster 5), choroid plexus (cluster 7), microglial (cluster 10) as well as PNS glial (cluster 12) identity, showed activated SOX2 target genes significantly upregulated. In contrast, progenitors with a dorsal forebrain regional identity (cluster 0) downregulated these genes (Fig. 5E).

Our data show that upon compression, there is an increase in SOX2 protein levels. Furthermore, we found functional implications to this with the induction of SOX2 target genes in compressed cells (Fig. 5E) correlating with changes in the lineage decisions of neural stem cells. These findings align with prior studies showing that SOX2 protein level changes comparable to those observed here promote neural stem cell proliferation (Bylund et al, 2003; Chew et al, 2005; Graham et al, 2003; Kopp et al, 2008).

Next, we asked which molecular processes are affected by mechanical manipulation in neural stem and progenitor cells. To this end, we performed a GO enrichment analysis of the differentially expressed genes between mock and compressed neural progenitors (clusters 0, 1, 5, 9) and found miRNA processing as the strongest enriched GO term. SOX2 protein levels are strongly regulated by posttranscriptional and posttranslational regulatory networks (Zhang et al, 2020) with miRNAs being central to fine-tune the exact SOX2 protein levels (Mirzaei et al, 2021). Interestingly, mechanical stimuli such as matrix stiffness or cyclic stretch can influence the expression and activity of mechanosensitive miRNAs, linking biophysical forces to post-transcriptional gene regulatory networks that ultimately modulate levels of target proteins (summarized in reviews of mechanosensitive miRNAs) (Chen et al, 2017; Herault et al, 2021). miRNAs such as the miR-200 family and miR-145 directly target the untranslated region (UTR) of SOX2 mRNA, repressing its translation and reducing SOX2 protein levels in stem cells and cancer cells (Chuang et al, 2020; Li and He, 2012; Mirzaei et al, 2021; Sun et al, 2013) thereby controlling proliferation and differentiation. Mechanotransduction studies have shown that mechanical forces can up- or down-regulate specific miRNAs via focal adhesions, the cytoskeleton and specific signaling pathways (Frith et al, 2018) suggesting a plausible route for mechanical forces to control SOX2 through mechanosensitive miRNA modulation. Together, these findings provide a possible molecular explanation how mechanical manipulations could regulate SOX2 protein levels post-transcriptionally

through the modulation of miRNAs that bind to SOX2 mRNA and regulate its translation.

In agreement with the bulk RNA-seq data, we found an enrichment of terms associated with cellular metabolism such as mitochondrial electron transport and mitochondrial ATP synthesis and signaling pathways implicated in patterning of the early nervous system such as WNT signaling (Fig. 5F). However, leveraging the power of scRNA-seq we next stratified the data along the diffusion pseudotime (Fig. EV3L), thereby being able to address molecular changes along differentiation trajectories. This revealed an overall downregulation of the GO term comprising genes associated with mitochondrial ATP synthesis in compressed cells, while the GO term associated with another metabolic pathway such as regulation of cholesterol biosynthetic process (Fig. 5G) was increased particularly late along pseudotime, highlighting a stage-specific effect on cellular metabolism along differentiation trajectories of neural lineages.

We furthermore analyzed these patterning and metabolism-associated GO terms prioritized by the bulk RNA-seq dataset (Fig. 4C) and assessed the difference of the expression of the respective GO term associated genes across the neural stem and progenitor and neuron clusters between mock and 50% compressed samples. This revealed that the impact of mechanical manipulation on the expression of these genes is lineage and cluster-dependent. The GO terms 'pattern specification processes' and 'mitochondrial ATP synthesis coupled electron transport' are deregulated in most clusters (Fig. EV3M,N). On the other hand, lipid metabolism-associated GO terms show a more cluster specific changes (Fig. EV3O), deregulated in neuronal clusters. Hence, the compression induced changed cellular behavior of neural stem and progenitor cells directly correlates with changes in mitochondrial ATP synthesis in neural stem and progenitor cells while the changes in lipid metabolism are rather associated with neuronal cell types.

Together, these data uncovered an uncharted level of regulation of neural stem cells by mechanics early during development. We have shown that acute as well as persistent mechanical manipulations alter neural stem cell lineage decisions by tipping the balance between proliferation and differentiation of neural stem and progenitor cells thereby decreasing neurogenesis. Moreover, our data illustrate that mechanical manipulations govern the transcriptional networks associated with early patterning events establishing the dorso-ventral as well as the anterior-posterior axis. Interestingly, these effects vary across cell types with different regional identities

suggesting a cell type-specific susceptibility to mechanical impacts. The nature of the molecules involved in the transduction of the mechanical stimuli to cellular behavior is yet unknown. While there are many potential candidate mechanisms (Pillai and Franze, 2024), involvement of mechanosensitive ion channels allowing an influx of cations such as $Na^+$, $K^+$ and $Ca^{2+}$ upon mechanical stimuli is in agreement with the bulk RNA-seq data as Calcium signaling pathway associated genes show deregulation upon mechanical manipulation (Fig. 4E). Moreover, mechanical forces directly impact on microtubule dynamics (Brouhard and Rice, 2018), a process which in turn orchestrates neural stem cell behavior regulating the balance between proliferation and differentiation (Camargo Ortega et al, 2019). The contribution of the various different mechanotransduction mechanisms remains to be shown. Moreover, it would be interesting to design future experiments to address whether and how both acute compression and a persistent change in the mechanical environment impact brain organoids of a defined regional identity, i.e., dorsal or ventral brain organoids generated through guided protocols (Paşca et al, 2022).

# Methods

### Reagents and tools table

| Reagent/Resource | Reference or Source | Identifier or Catalog Number |
|---|---|---|
| **Experimental models** | | |
| HMGUi001-A (CVCL_WJ49), Human induced pluripotent stem cell (hiPSC), female, Caucasian | Helmholtz Zentrum Munchen; Neuherberg; Germany | https://hpscreg.eu/cell-line/HMGUi001-A |
| HPSI0214i-wibj_1, Human induced pluripotent stem cell (hiPSC), female, Caucasian | Wellcome Sanger Institute (WTSI), Hinxton, Cambridge, United Kingdom | https://ebisc.org/WTSIi046-B |
| HPSI0314i-hoik_1, Human induced pluripotent stem cell (hiPSC), female, Caucasian | Wellcome Sanger Institute (WTSI), Hinxton, Cambridge, United Kingdom | https://ebisc.org/WTSIi026-A |
| **Recombinant DNA** | | |
| **Antibodies** | | |
| Anti-MAP2 mouse monoclonal, 1:300 | Sigma/Aldrich (Merck) | M4403 |
| Sox2 (D6D9) rabbit, monoclonal, 1:300 | Cell Signaling Technology | 3579 |
| Anti-NeuN Antibody, clone A60, mouse, monoclonal, 1:300 | Merck Millipore | MAB377 |
| Goat anti-Rat IgG (H + L) Cross-Adsorbed Secondary Antibody, Cyanine3, Polyclonal, (1:1000) | Thermofisher (Invitrogen) | A21121 |
| Goat anti-Rabbit IgG (H + L) Cross-Adsorbed Secondary Antibody, Alexa Fluor™ 488, polyclonal, (1:1000) | Thermofisher (Invitrogen) | A11008 |

| Reagent/Resource | Reference or Source | Identifier or Catalog Number |
|---|---|---|
| Goat anti-Mouse IgG1 Cross-Adsorbed Secondary Antibody, Alexa Fluor™ 647, polyclonal, (1:1000) | Thermofisher | A-21240 |
| Anti-BrdU antibody [BU1/75 (ICR1)] - Proliferation Marker, rat, monoclonal, 1:300 | Abcam | ab6326 |
| Anti-Cleaved Caspase-3 antibody [E83-77], rabbit, monoclonal, 1:300 | Abcam | ab32042 |
| **Oligonucleotides and other sequence-based reagents** | | |
| **Chemicals, Enzymes and other reagents** | | |
| 2-Mercaptoethanol | Merck/Sigma | M3148 |
| 6-well plates | 6 Greiner Bio-One | 657160 |
| Accutase | Life Technologies | A1110501 |
| Adhesive slide Superfrost® Plus | Roth | H867.1 |
| Aqua-Poly/mount | Polysciences | 18606-20 |
| B27 - Vitamin A Supplement | Life Technologies | 12587-010 |
| B27 + Vitamin A Supplement | Life Technologies | 17504-044 |
| bFGF (FGF2) | Peprotech | 100-18B |
| DAPI, 4',6-Diamidine-2'-phenylindole dihydrochloride | Merck | 10236276001 |
| DMEM/F12 + GlutaMAX (1X) | Life Technologies | 31331-028 |
| Dynabeads™ MyOne™ SILANE | Invitrogen | 37002D |
| Ethanol absolte ACS, Ph. Eur, | Sigma | 1009831011 |
| Gelatin, Type A, from porcine skin, gel strength 300 bloom | Provided by Aldo R. Boccaccini's group, FAU Erlangen, Germany | https://doi.org/10.1089/ten.tec.2022.000 and https://doi.org/10.1016/j.actbio.2025.09.007 |
| Glycerol 100% | Sigma | SBL3980 |
| Goat Serum | Thermofisher (Gibco) | 16210064 |
| Heparin | Sigma-Aldrich | H3149 |
| hESC-quality FBS | Life Technologies | 10270-106 |
| HistoVT One (1:10 in distilled water) | Gerbu Biotechnik | 06380 |
| Insulin solution | Sigma-Aldrich | I9278-5ML |
| Knockout serum replacement KOSR | Life Technologies | 10828-028 |
| Matrigel for embedding | Corning | 354234 |
| Matrigel reduced growth factor for cell plate coating | Corning | 354230 |
| MEM-NEAA | Life Technologies | 11140050 |

| Reagent/Resource | Reference or Source | Identifier or Catalog Number |
|---|---|---|
| Microbial transglutaminase | Provided by Aldo R. Boccaccini's group, FAU Erlangen, Germany | https://doi.org/10.1089/ten.tec.2022.000 and https://doi.org/10.1016/j.actbio.2025.09.007 |
| Millex-VV Steril filters with Luer Lock system, PVDF, 0.10 μm, Ø 33 mm | Merck | SLVV033RS |
| mTeSR Plus Kit | STEM Cell Technologies | 100-0276 |
| N2 supplement | Life Technologies | 17502-048 |
| Neurobasal (1X) | Life Technologies | 21103049 |
| Nuclease-free water | Invitrogen | AM9937 |
| Oxidized hyaluronic acid | Synthetized and provided by Aldo R. Boccaccini's group, FAU Erlangen, Germany | https://doi.org/10.1089/ten.tec.2022.000 and https://doi.org/10.1016/j.actbio.2025.09.007 |
| Paraformaldehyde (PFA), 4% Solution in PBS (with DEPC) | Booster Bio/BioTrend | AR1068 |
| PBS | Sigma-Aldrich | P4417 |
| PBS without MgCl$_2$, CaCl$_2$ | Sigma-Aldrich | 2687504 |
| Pen/Strep | Thermo Fisher Scientific | 15140-122 |
| Qiagen Buffer EB | Qiagen | 172044351 |
| qpore® Steril filters with Luer Lock system, PES, steril, 0.45 μm, Ø 33 mm | NeoLab | 60069 |
| Rock Inhibitor Y27632 | STEM Cell Technologies | 72304 |
| Sterile syringe Terumo with Luer Lock system: 10 ml | Fisher Scientific | 11547302 |
| Triton™ X-100 | Merck | 9036-19-5 |
| **Software** | | |
| Agilent 2100 Expert Software | Agilent | B.02.11.SI824 (SR1) |
| Python 3.8, 3.10 and 3.12 | Python Software Foundation | https://www.python.org/ |
| **Other** | | |
| 10 cm Petri dishes | Corning | CLS430591 |
| 10x Magnetic Separator | 10x Genomics | 120250 |
| 10x Vortex Adapter | 10x Genomics | 120251 |
| 96-well ultra-low attachment plates | Corning | CLS7007 |
| Agilent Bioanalyzer, G2939B | Agilent | DEDAE02639 |
| Centrifuge 5810 | Eppendorf | 10595074 |
| Bioanalyzer High Sensitivity DNA Analysis Kit | Agilent | 5067-4626 |
| Cell countess, Countess II FL | Invitrogen | Serial No.2186A19105460 |
| HERASAFE 2030i, Cell culture hood | Thermo Scientific | |

| Reagent/Resource | Reference or Source | Identifier or Catalog Number |
|---|---|---|
| Integra vacusafe, Cell culture pump | Integra | 158310 |
| Chromium Next GEM Chip G Single Cell Kit, 16 rxns | 10x Genomics | PN-1000127 |
| Chromium Next GEM Chip G Single Cell Kit, 48 rxns | 10x Genomics | PN-1000120 |
| Chromium Next GEM Secondary Holder | 10x Genomics | 1000195 |
| Chromium Next GEM Single Cell 3′ GEM, Library & Gel Bead Kit v3.1, 16 rxns PN-1000121 | 10x Genomics | PN-1000121 |
| Chromium Next GEM Single Cell 3′ GEM, Library & Gel Bead Kit v3.1, 4 rxns | 10x Genomics | PN-1000128 |
| Chromium Next GEM Single Cell 3′ Reagent Kits v3.1 | 10x Genomics | User guide |
| Cryostar NX70 | Thermo Fisher Cryostar NX70 | 15320755 |
| Evos XL Core, Microscope-cell culture | Invitrogen | AMEX1000 |
| EVOSTM Imaging System, M7000 | Thermo Fisher | AMF7000HCA |
| High Sensitivity DNA Reagents | Agilent | 5067-4627 |
| Incubator | Panasonic | MCO-170AICUVD-PE |
| Neural dissociation Kit P | Miltenyi Biotec | 130-092-628 |
| C1000 Touch-thermal cycler (PCR machine) | Bio-Rad | Serial No. CT040169 |
| PCR Tube strips 0.2 ml | Eppendorf | L208220R, 1302 |
| ROCKER 3D digital (Plate shaker) | IKA | 0004001000 |
| Rainin 200 μl bioclean ultra wide-O, LR filter | Rainin | PN 30389241 |
| Rainin Pipet-Lite XLS 20-200 μl | Rainin | 17014410 |
| Single Index Kit T Set A, 96 rxns | 10x Genomics | PN-1000213 |
| Thermomixer compact F1.5 | Eppendorf | 5384000020 |
| Vortex Genie 2 | Scientific Industries | 2E-244519 |

## Culture of human iPSC lines and generation of brain organoids

Throughout the study, we used the hiPSC lines wibj2 and hoik1 from the HipSci feeder free panel (ECACC 77659901) as well as XM001 hiPSCs (https://hpscreg.eu/cell-line/HMGUi001-A) (Wang et al, 2018) which are derived from fibroblasts of a healthy female and were obtained from Heiko Lickert (Helmholtz Center Munich, Germany). The XM001 hiPSC line constitutively expressed the

fluorescent DsRed protein, which was achieved by PiggyBac transposon integration as described earlier (Menon et al, 2023). All cells were regularly tested for the presence of mycoplasma using the PCR Mycoplasm Test Kit (PromoKine) or the LookOut Mycoplasma PCR detection kit (Sigma-Aldrich). mTeSR Plus medium (STEMCELL Technologies) was used to culture all hiPSC lines used in this study. Cells were grown on Matrigel-coated 6-well plates in 5% $CO_2$ at 37 °C until a confluency of 80–90% was reached. Brain organoid formation was conducted using a published protocol including small adaptations (Lancaster et al, 2013). Briefly, accutase (Thermo Fisher) was used to generate single-cell suspensions of hiPSCs. Following centrifugation, cells were resuspended in organoid formation medium (OFM) supplied with 4 ng/ml of low bFGF (Peprotech) and 5 µM ROCK-inhibitor Y-27632 (STEMCELL Technologies). OFM consisted of DMEM/F12 + GlutaMAX-I (Thermo Fisher), 20% KOSR (Thermo Fisher), 3% FBS (Thermo Fisher), 0.1 mM MEM-NEAA (Thermo Fisher), 0.1 mM 2-mercaptoethanol (Sigma-Aldrich). 9000 cells in 150 µL OFM/well were aggregated in low attachment 96-well plates (Corning) for at least 48 h during which embryoid bodies (EBs) formed. After 72 h half of the medium was replaced with 150 µl of new OFM without bFGF and ROCK-inhibitor. At day 5 neural induction medium (NIM) consisting of DMEM/F12 + GlutaMAX-I (Gibco), 1% N2 supplement (Gibco), 0.1 mM MEM-NEAA (Gibco), and 1 µg/ml Heparin (Sigma-Aldrich) was added to the EBs in the 96-well plate to promote their growth and neural differentiation. NIM was changed every two days until day 12/13, when aggregates were transferred to undiluted Matrigel (Corning) droplets. The embedded organoids were transferred to a petri dish (Greiner Bio-One) containing organoid differentiation medium (ODM) without vitamin A. Three or four days later, the medium was exchanged with ODM with vitamin A and the plates were transferred to an orbital shaker set to 30 rpm inside the incubator. Medium was changed twice per week. ODM consisted of a 1:1 mix of DMEM/F12 + GlutaMAX-I (Thermo Fisher) and Neurobasal medium (Thermo Fisher), 0.5% N2 supplement (Thermo Fisher), 0.1 mM MEM-NEAA (Thermo Fisher), 100 U/ml penicillin and 100 µg/ml streptomycin (Thermo Fisher), 1% B27 +/- vitamin A supplement (Thermo Fisher), 0.025% insulin (Sigma-Aldrich), 0.035% 2-mercaptoethanol (Sigma-Aldrich).

## Rheometer-mediated large-strain compression of brain organoids

We used a Discovery HR-3 rheometer from TA Instruments (New Castle, Delaware, USA) with a heated Peltier plate (37 °C) to measure the organoids' response under cyclic compression. After calibrating the instrument, experimental organoids were transferred to the rheometer platform using a pipette with cut 1000 pipette tip. Upon transfer to the platform, excess medium was removed. Before starting the experiment, the upper geometry of the rheometer was lowered manually until it contacted the organoid (Fig. 1A). The gap value recorded by the instrument upon contact was assumed to be the initial height X of the organoid and used to calculate the compressed height Y. Organoids were compressed to 40, 50, or 60% of their original height. Three consecutive cycles of compression loading and unloading were performed, each lasting of about 1 min. Gap values and forces were recorded throughout the entire experiment. The recorded data were analyzed using a custom-written MATLAB code. Mock organoids were placed on the rheometer platform for approximately one minute (i.e., the time it takes to compress) and transferred back to the medium without compression. After compression, organoids were transferred into organoid medium and placed back on the shaker for either 24 h or 5 days. Ctrl group organoids were not taken out from the incubator. Both, the mock and the ctrl organoids are jointly referred to as 'uncompressed'.

## Generation of hydrogels with defined physical properties

We employed hydrogels (HG) built from oxidized hyaluronic acid (OHA) and gelatin crosslinked by microbial transglutaminase. OHA was synthesized from hyaluronic acid powder by sodium periodate oxidation (Kuth et al, 2022). OHA/GEL HGs were produced as described previously (Kuth and Boccaccini, 2024) and used to encapsulate the organoids. In contrast to Matrigel, these HGs can be reproducibly fabricated and finetuned to a specific stiffness. We tested 1.25%OHA/2.5%GEL with 230 Pa initial stiffness (referred to as HG1) and 2.5%OHA/2.5%GEL (referred to as HG2) with 500 Pa initial stiffness.

## BrdU labeling of brain organoids

The thymidin analogon BrdU (5-Brom-2′-deoxyuridin; Sigma-Aldrich) was used to label cells that went through cell division after addition of BrdU. BrdU in a final concentration of 10 µM was added to the organoid medium of either compressed or uncompressed ctrl organoids right after the rheometer experiment for a duration of 24 h and analyzed at 5 days after compression. To correlate the physical forces which were applied through compression with the subsequent analyses, individual organoids were kept separated after compression until fixation.

## Organoid fixation and immunofluorescence

For fixation, organoids were transferred to 1.5 ml tubes. Organoids were washed with PBS and then fixed with 1xPBS buffered 4% paraformaldehyde (PFA, Carl Roth) for 30 min. Time of PFA fixation was extended up to one hour depending on the size of the organoids. Afterwards, organoids were washed three times for 10 min with PBS and incubated in 30% sucrose (Sigma-Aldrich) in PBS for cryoprotection. For cryosectioning, organoids were embedded in Neg-50™ Frozen Section Medium (Thermo Fisher) on dry ice. Frozen organoids were cryosectioned in 30-µm sections using the Thermo Fisher Cryostar NX70 cryostat. Sections were placed on SuperFrost Plus Object Slides (Thermo Fisher) and stored at −20 °C until use. For post-fixation, organoid slices were incubated with 4% PFA for 15 min followed by three washing steps with PBS for 5 min. HistoVT One (Fisher Scientific) antigen retrieval was performed, diluting HistoVT One 1:10 in distilled water and incubation of the organoid slices in this solution at 70 °C for 20 min. During the entire staining procedure, slides were kept in humidified staining chambers in the dark. Organoid slices were washed briefly with blocking solution (PBS, 4% normal donkey serum (NDS, Sigma-Aldrich), 0.25% Triton X-100 (Sigma-Aldrich)) followed by a 2-h incubation with blocking solution at room temperature. Primary antibodies were diluted in antibody solution (PBS, 4% NDS, 0.1% Triton X-100) and tissue sections

were incubated overnight at 4 °C. Next, following two washes using PBS for 5 min and one with PBS containing 0.5% Triton X-100 for 8 min, secondary antibodies were added diluted in antibody solution and incubated for 2 h at room temperature. Sections were washed three times with PBS for 5 min. Slides were counterstained with DAPI (Sigma-Aldrich) 1:1000 in PBS for 5 min, followed by one washing step with PBS. Lastly, organoid sections were mounted using Aqua Polymount (Polysciences). Antibodies used were selected according to the antibody validation reported by the distributing companies. Slides were stained with the following primary antibodies (1:300 dilution): mouse anti-MAP2 antibody (Sigma-Aldrich, M4403), Phalloidin-Atto 647N to target F-actin (Sigma-Aldrich, 65906-10NMOL), rabbit anti-SOX2 (Abcam; ab137385; 1:300), anti-BrdU (Abcam; ab6326), NEUN (Merck Millipore, MAB377), rabbit anti-cCAS3 (Abcam; E8377, 1:300). The following secondary antibodies were used (1:1000 dilution): goat anti-mouse IgG1 Alexa 555 (Thermo Fisher, A21127), goat anti-rabbit Alexa 647 (Thermo Fisher, A21245), goat anti-rabbit Alexa Cy3 (Thermo Fisher, A10520), goat anti-mouse IgG1 Alexa 647 (Thermo Fisher, A21240), goat anti-rat Alexa 555 (Thermo Fisher, A10522). For staining SOX2, NEUN, and BrdU together, we used serial staining with 2 N HCl treatment to denature DNA for 20 min at room temperature, followed by neutralization with 0.1 Na-tetraborate (pH 8.5) two times for 15 min. After washing with PBS, Histo VT one was as described above. First, the stainings against SOX2 and NEUN were performed. Subsequently BrdU was stained.

## Microscopy and image analysis

Epifluorescence pictures were taken using the EVOS M7000 Imaging System (Thermo Fisher). Z-stacks of the BrdU images were taken using an Apotome.2 (Zeiss) equipped with the Colibri 5 light source (Zeiss). Images were analyzed using FIJI (v1.52-1.53).

Segmentation of organoid pictures into ventricular zone-like structures (VZLS) and neural compartments (NC) was done as previously described (Frank et al, 2024). In short, the freehand selection tool of FIJI (FIJI v1.52-1.53) was used to delineate the outer organoid border as well as the apical and basal sites of individual VZLS based on the DAPI channel. To quantify SOX2 levels in organoids we used a custom phyton-based script employing basic python (v3.8-3.9) functions and opencv (4.5.0-4.10.0) to measure the pixel values of the SOX2 channel in each individual pixel of a picture and annotate each pixel with its corresponding compartment. To measure the distance for each pixel of an organoid picture to the outer boundary, the apical as well as the basal side of the corresponding VZLS we used the cKDTree function of the scipy (v1.10.1) package. To visualize the density distribution of SOX2 levels in VZLS we employed the seaborn (v. 0.13.2) kdeplot function.

## Bulk RNA-seq data processing and analysis

For bulk RNA-seq we used d30 brain organoids which were either compressed 50% or 60% and mock as well as ctrl organoids 24 h after compression. Individual organoids were processed by lysing in RLT buffer containing beta-mercaptoethanol (Sigma-Aldrich). Total RNA was isolated using the RNeasy Mini Kit (Qiagen) and a Bioanalyzer (Agilent Technologies) was used for quantification

and quality control of RNA samples. Poly-A enrichment-based library preparation and transcriptome sequencing was performed by Novogene Europe (Cambridge, GBLibraries). Paired-end 150 bp reads were sequenced on a NovaSeq6000.

The FASTQ files were preprocessed using fastp (v0.23.2) (Chen et al, 2018) with the following settings: qualified_quality_phred 20, unqualified_percent_limit 10, n_base_limit 2, length_required 20, low_complexity_filter enable, complexity_threshold 20, dedup enable, dup_calc_accuracy 6, overrepresentation_analysis, detect_adapter_-for_pe, cut_right). The resulting FASTQ files were aligned to the human genome GRCH38 release 42 from GENCODE using R (v4.1.3) and the R package Rsubread (v2.8.2) (Liao et al, 2019). To count the aligned reads, we employed the Rsubread built-in function *feature_-count* with the following settings: (isPairedEnd=T, countReadPairs=T, requireBothEndsMapped=T, countChimericFragments=T, countMultiMappingReads=F, allowMultiOverlap=F). We then used EdgeR (Robinson et al, 2010) (v4.4.0) to filter genes using the EdgeR built-in function *filterByExpr* with default values and normalized differences in sequencing depth between samples employing *calcNormFactors*. PCA analysis was performed with EdgeR and plotted using ggplot2 (v3.5.1). For the confidence ellipses, we took advantage of the R package ggpubr (v0.6.0). Differential gene expression was calculated using EdgeRs *glmQLFIT* and *glmTreat* functions. Genes with a *p*-value < 0.01 a FDR < 0.01 and a log2FC > 2 were considered as differentially expressed in the respective comparisons. The upset plots showing the overlap of differential expressed genes between different conditions were produced using the R package VennDetail (v.1.21.0). GO analysis was performed using topGO (v.2.52.0) employing the classic algorithm with Fishers exact test for significance testing. We considered GO terms with a *p*-value < 0.05 and having at least 5 differentially expressed genes per GO term. The top 15 enriched GO terms are shown. Heatmaps show the z-score of the mean per condition of the normalized, log2 transformed CPM using the R package pheatmap (v.1.0.12). To calculate the transcriptional deviation of KEGG pathways we employed the R packages KEGGREST (v.1.46.0) to retrieve the gene names associated with a pathway. We then calculate for each gene in a KEGG pathway in each group (mock, ctrl, 50% compression, 60% compression) the natural logarithm of the fold change over the mean in the ctrl samples (mock and ctrl), summed the results up for all genes in a KEGG pathway and normalized the result by dividing with the number of annotated genes present in our dataset. The *p*-values were calculated using one-way ANOVA testing employing the R function aov() and TukeyHSD for Tukey's post hoc test. Results were plotted using the R package pheatmap (v.1.0.12).x.

## Single-cell RNA-seq data generation

For the scRNA-seq experiment, organoids of two different hiPSC lines (wibj2, XM001) were used. For each hiPSC line we used a pool of 3 mock organoids as controls and 2 individual compressed organoids, resulting in 4 compressed samples and 2 mock samples in total. The organoids dissociated 24 h after 50% compression using the Neural Tissue Dissociation Kit P (Miltenyi Biotec). Briefly, selected organoids were cut into smaller pieces, washed with medium, and subsequently three times for 5 min with 1xPBS. Organoid pieces were transferred to a tube containing the enzyme mix P (according to the manufacturer's protocol) and incubated at 37 °C for 10 min. Pieces were then triturated gently with a 1000p

pipette tip and incubated for another 10 min at 37 °C in the presence of enzyme mix A (according to the manufacturer's protocol). Pieces were then triturated gently with a 1000p and a 200p pipette tip and incubated for 5 min at 37 °C. Cell suspension was filtered using a 30 μm filter (Miltenyi Biotec) and centrifuged at $300 \times g$ for 5 min. After a second filtration step employing a 20 μm filter (Miltenyi Biotec) and subsequent centrifugation as described above, the cell pellet was resuspended in 100 μl 1xPBS (without $Ca^{2+}$ and $Mg^{2+}$). Cells were counted and tested for viability with Trypan Blue and the automated cell counter Countess (Thermo Fisher). Before loading the cells on the 10X Genomics Next GEM chip G v3 one sample of the wibj2 and one sample of the XM001 line were pooled. Cells were then loaded to obtain approximately 4000 cells per lane. Libraries were constructed according to the protocol of 10X genomics and sequenced together on an Illumina NovaSeq 6000 at the Genomics Core Facility of the Helmholtz Zentrum Munich.

### Single-cell RNA-seq data preprocessing, clustering, visualization

The functions count and aggr of the Cell Ranger software (10x Genomics. v6.0.2) were used for aligning to the GRCh38 reference genome and for sequencing depth normalization. Demultiplexing of the samples was done based on the RFP expression in the XM001 line. Scanpy (Wolf et al, 2018) (v.1.9.6-1.10.4) was used for further preprocessing, clustering, embedding, and visualization. Cells were excluded in which we detected less than 2500 genes, less than 5000 UMI counts, more than 25,000 counts, fraction of the transcriptome more than 7% mitochondrial genes. Moreover, genes expressed in less than 10 cells were excluded from further analysis. Doublets were identified with Scrublet (Wolock et al, 2019) and filtered out. After normalization and log transformation, highly variable genes were determined using the default settings in Scanpy. Cell cycle scores, percentage of mitochondrial genes, and number of detected UMIs were regressed out to reduce their confounding effects. PCA analysis with the arpack wrapper in SciPy (v1.7.0) followed by determining the 10 closest neighbors in the top 20 PCs and force-directed graph embedding utilizing ForceAtlas2 (Jacomy et al, 2014) provided by the python package fa2 (v.3.5) was used for embedding the transcriptomes. Clustering was performed with Scanpy's python implementation of the Leiden (Traag et al, 2019) algorithm (v.0.8.7). Developmental pseudotime was calculated using the diffusion pseudotime implementation (Haghverdi et al, 2016) in scanpy on the first 15 PCs setting the root cell in cluster 1.

### Differential gene expression and GO term analysis

Differentially expressed genes in progenitors (leiden clusters 0, 1, 5, 9) was determined between compressed and mock cells using the rank_genes_groups function of Scanpy and testing for significance with the Wilcoxon rank-sum test. Genes with an adjusted $P$-value < 0.01 were considered differentially expressed. GO analysis was performed using topGO (v.2.57.0) employing the default weight01 algorithm with Fishers exact test for significance testing. We considered GO terms with a $p$-value < 0.05 and having at least 5 differentially expressed genes per GO term. The data were visualized using seaborn (v. 0.13.2).

### SOX2 activated targets

The target genes of SOX2 were derived from the CollecTRI network (Müller-Dott et al, 2023) through the python packages decoupler (v.1.7.0) and omnipath (v.1.0.8) and only genes annotated as activated by SOX2 were considered. The scanpy function score_genes was used to calculate the transcriptional activity of the SOX2 target genes in each cell of the dataset. To calculate the fold change of the activity of the SOX2 target genes in each leiden cluster, the SOX2 score in each cell was normalized to the mean of the uncompressed cells in that cluster and the mean in each cluster and condition was calculated. Significance was tested employing the Man-Whitney U test implementation of scipy (v.1.7.0).

### GO-terms over pseudotime

To retrieve the genes associated with a given GO term we used the biomaRt (v.2.61.3) library in R. Using the scanpy function score_genes, we then calculated an expression score for each GO-term and stratified the expression of the GO-term over the diffusion pseudotime. The data was visualized using the regplot function in seaborn (v. 0.13.2).

### Statistics and reproducibility

Data were analyzed with R or Phyton using statistical tests indicated throughout the manuscript. No statistical methods were used to predetermine sample size. The investigators were not blinded during the analysis of the data. The experiments were not randomized.

## Data availability

The FASTQ files of bulk and the scRNA-seq data are deposited in the European Nucleotide Archive (ENA) at EMBL-EBI under accession number PRJEB103796. The data that support the findings of this study are available from the corresponding authors upon reasonable request.

The source data of this paper are collected in the following database record: biostudies:S-SCDT-10_1038-S44319-026-00719-2.

## Peer review information

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

## Acknowledgements

We thank Michael Wegner (FAU Erlangen-Nürnberg) for sharing equipment and lab space. We thank the Genomics Core Facility of the Helmholtz Zentrum Munich for their support. This work was supported by grants from the Bavarian State Ministry of Sciences, Research, and the Arts (ForInter; F.2-F2412.30/1/24) to MK and SF, from the Interdisciplinary Center for Clinical Research (IZKF) at the University Hospital of the FAU Erlangen-Nürnberg to MK (P068; Jochen-Kalden funding programme N7, S1 Train) and SF (P074, E32), and from the German Research Foundation (565201020) to SF and (426809158; GRK2162/2 TP C2) to MK, as well as German Research Foundation (DFG) project 460333672 CRC1540 EBM to SB, AB, MK, and SF.

## Author contributions

**Hanna Lampersperger**: Investigation; Writing—review and editing. **Michael Tranchina**: Formal analysis; Investigation; Writing—review and editing. **Bastian Meth**: Investigation; Writing—review and editing. **Dandan Han**: Investigation; Writing—review and editing. **Negar Nayebzadeh**: Investigation; Writing—review and editing. **Nina Reiter**: Investigation; Writing—review and editing. **Sonja Kuth**: Investigation; Writing—review and editing. **Markus Lorke**: Investigation; Writing—review and editing. **Aldo R Boccaccini**: Supervision; Methodology; Writing—review and editing. **Silvia Budday**: Supervision; Funding acquisition; Methodology; Writing—review and editing. **Marisa Karow**: Conceptualization; Supervision; Funding acquisition; Visualization; Writing—original draft; Project administration; Writing—review and editing. **Sven Falk**: Conceptualization; Data curation; Formal analysis; Supervision; Funding acquisition; Visualization; Writing—original draft; Project administration; Writing—review and editing.

Source data underlying figure panels in this paper may have individual authorship assigned. Where available, figure panel/source data authorship is listed in the following database record: biostudies:S-SCDT-10_1038-S44319-026-00719-2.

## Funding

## Disclosure and competing interests statement

The authors declare no competing interests.

# Expanded View Figures

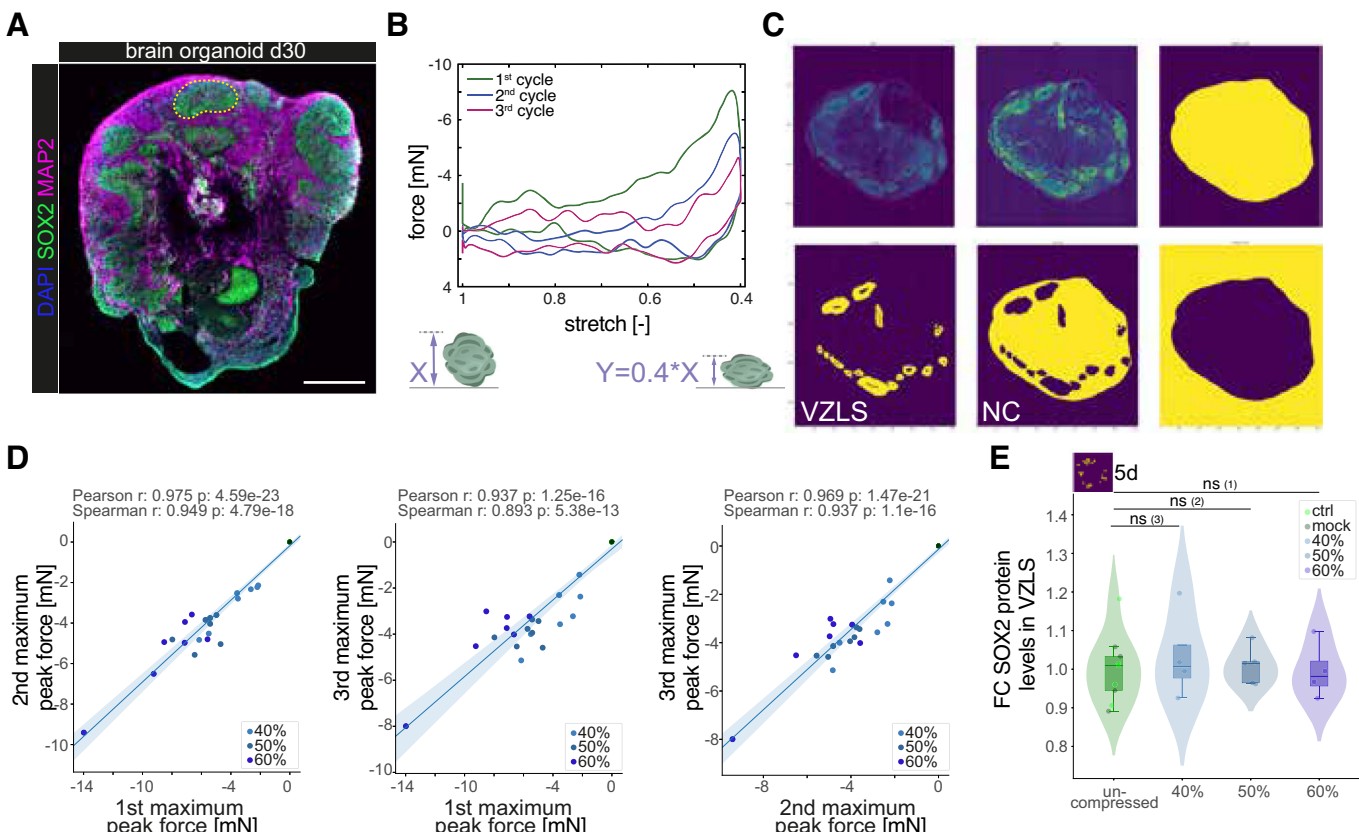

**Figure EV1. Acute compression of brain organoids results in transient SOX2 upregulation.**

(**A**) Immunofluorescence images of a section from a d30 brain organoid stained for SOX2 (green), MAP2 (magenta), and DAPI (nuclei). Dashed line highlights a VZLS containing SOX2 positive neural stem and progenitor cells, surrounded by MAP2 positive neurons. Scale bar = 500 μm. (**B**) Stretch-force curves measured during cycling compression of a d30 brain organoid. The x-axis shows the degree of deformation (1: no deformation. 0.4: 60% compression). The y-axis shows the applied force. (**C**) Example segmentation of a d30 organoids into ventricular zone-like structures (VZLS) and neuronal compartment (NC) regions. (**D**) Linear regression showing the correlation between the maximum peak force of the first compression cycle and the second cycle (left plot), the first and the third cycle (middle plot), and the second and the third cycle (right plot) across conditions. Pearson as well as Spearman correlations are shown in the plots. Dots represent individual compressed organoids (d30). (**E**) Violin, box, and jitter plots showing the fold change (FC) in mean SOX2 signal in SOX2-positive cells in the VZLS 5 days after compression, relative to uncompressed organoids, based on IF intensity measurements. Note no significant changes in SOX2 levels. Ctrl: $n = 5$; mock: $n = 4$; 40%: $n = 4$; 50%: $n = 5$; 60%: $n = 4$; Exact $P$ values: (1): 1.000; (2): $5.185 \times 10^{-1}$; (3): $7.105 \times 10^{-1}$. Violin plots show the distribution of the data. Boxplots show the median (center line), interquartile range (box), and whiskers extending to the most extreme values within 1.5× the interquartile range; dots represent individual organoids. Statistical significance was assessed using a two-sided Wilcoxon rank-sum test (ns: $P > 0.05$; *: $P < 0.05$; **: $P < 0.01$; ***: $P < 0.001$). Yellow/magenta images in the upper left corner indicate whether VZLS and/or NC were included (according to the segmentation shown in Fig. EV1C).

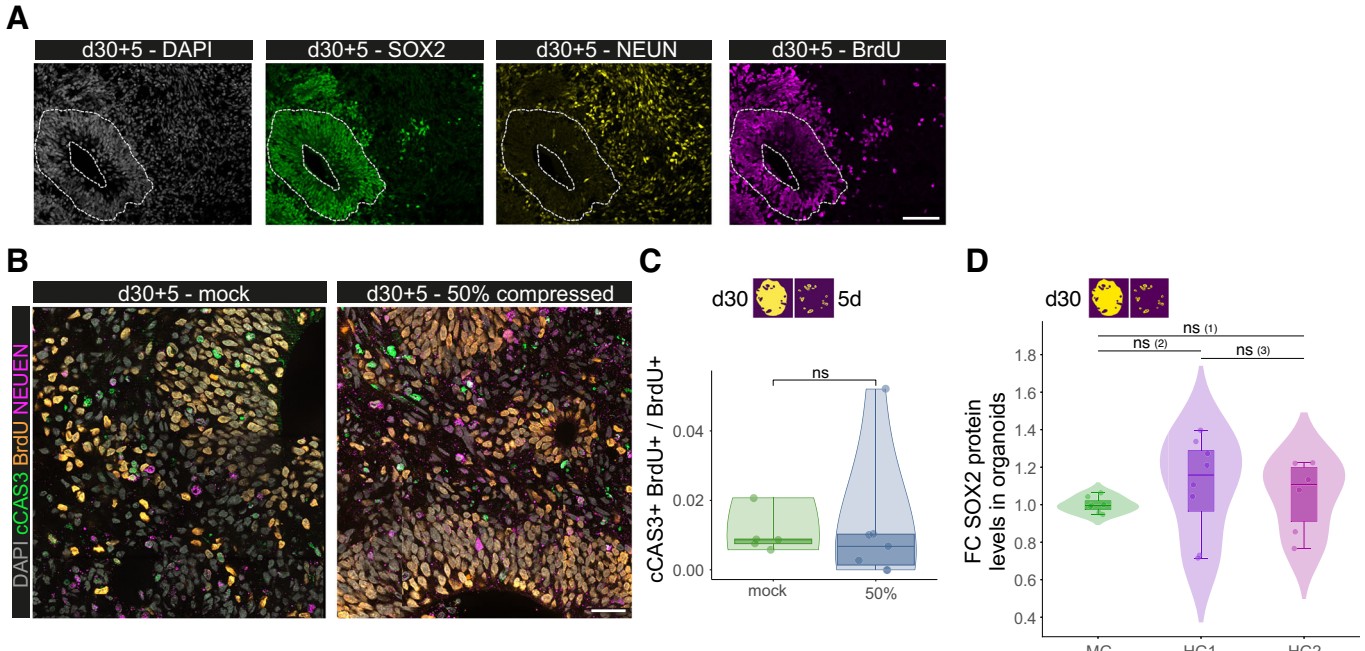

**Figure EV2.  Long-lasting changes on molecular and cell fate level.**

Immunofluorescence images of section from d30 organoids 5 days after compression and BrdU addition showing co-staining of the stem cell marker SOX2, the neuronal marker NEUN and the incorporation of BrdU (experimental paradigm: Fig. 4B). Dashed line delineates a VZLS. Scale bar = 100 μm. (**B**) Immunofluorescence images showing BrdU, cCAS3, NEUN co-staining in mock control (left image) and 50% compressed d30 organoids (right image) 5 days after compression and BrdU addition. DAPI was used to counterstain for nuclei. Scale bar = 50 μm. (**C**) Violin, box, and jitter plots showing the fraction of cleaved caspase 3 (cCAS3)/BrdU double positive cells among all BrdU positive cells to reveal the fraction of apoptotic cells among the cells born after compression. Mock: $n = 5$, 50%: $n = 7$; exact P value: 0.63. (**D**) Violin, box, and jitter plots showing the fold change (FC) in mean SOX2 signal in SOX2-positive cells in whole organoid sections of d30 organoids embedded in different matrixes, relative to MG embedded organoids, based on IF intensity measurements. MG = matrigel; HG1 = 1.25% OHA/2.5% GEL; HG2 = 2.5% OHA/2.5% GEL; dots represent individual organoids. MG: $n = 6$; HG1: $n = 8$; HG2: $n = 6$. Exact P values: (1): $1.520 \times 10^{-1}$; (2): $3.660 \times 10^{-1}$; (3): $7.546 \times 10^{-1}$. In (**C, D**), Violin plots show the distribution of the data. Boxplots show the median (center line), interquartile range (box), and whiskers extending to the most extreme values within 1.5× the interquartile range; dots represent individual organoids. Statistical significance was assessed using a two-sided Wilcoxon rank-sum test (ns: $P > 0.05$). Yellow/magenta images in the upper left corner indicate whether VZLS and/or NC were included (according to the segmentation shown in Fig. EV1C).

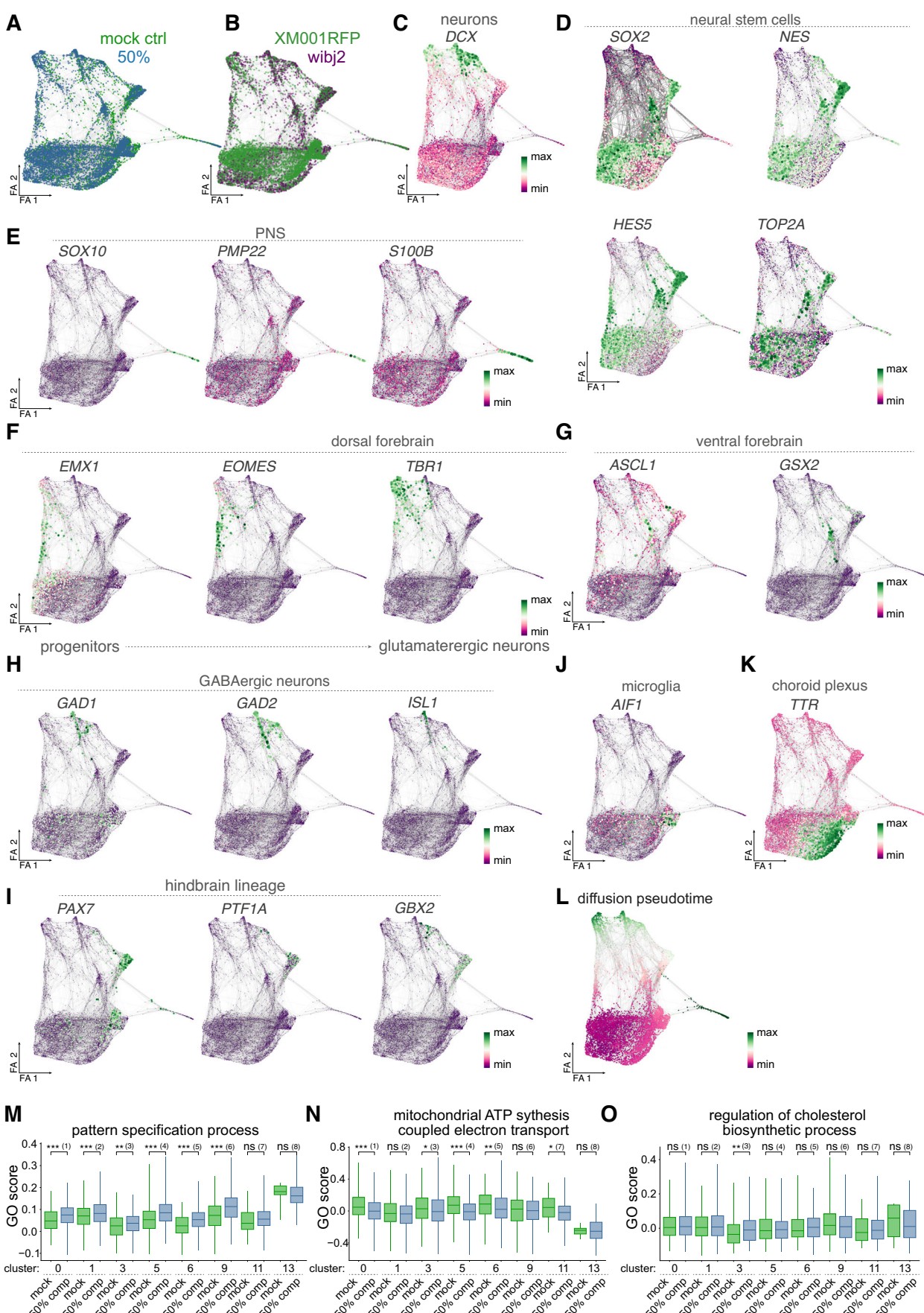

◀ **Figure EV3. Molecular dissection of the impact of compression on single-cell level.**

(A) force-directed graph embedding showing the contribution of mock and compressed cells, as well as (B) of the two different hiPSC lines XM001RFP and wibj2 used. Force-directed graph embedding showing (C) expression of the neuronal marker *DCX*, and (D) expression of the neural stem cell marker genes *SOX2, NES, HES5* and the proliferation marker *TOP2A*. (E) Expression of PNS markers (*SOX10, PMP22,* and *S100B*). (F) Expression of dorsal forebrain trajectory markers with the neural stem and progenitor marker *EMX1, EOMES* marking intermediate progenitor and the neuronal marker *TBR1*. (G) Expression of markers for ventral forebrain lineage cells (*ASCL1, GSX2*) and (H) GABAergic neurons (*GAD1, GAD2, ISL1*). (I) Expression of hindbrain lineage markers (*PAX7, PTF1A, GBX2*). (J) *AIF1* expression highlighting the microglia cluster. (K) Expression of choroid plexus marker (*TTR*). (L) Diffusion pseudotime plotted on the force-directed graph embedding. (A–I) FA refers to force atlas. The size of the dots indicates the extent of expression. (M) Boxplot showing the expression score of the GO term 'pattern specification process' within specific clusters comparing mock and 50% compressed samples, respectively. Exact $P$ values: (1): $4.8 \times 10^{-10}$; (2): $2.0 \times 10^{-4}$; (3): $6.5 \times 10^{-3}$; (4): $6.8 \times 10^{-11}$; (5): $2.5 \times 10^{-6}$; (6): $4.2 \times 10^{-8}$; (7): 0.086; (8): 0.69. (N) Boxplot showing the expression score of the GO term 'mitochondrial ATP synthesis coupled electron transport' within specific clusters comparing mock and 50% compressed samples, respectively. Exact $P$ values: (1): $3.4 \times 10^{-6}$; (2): 0.076; (3): 0.045; (4): $4.6 \times 10^{-9}$; (5): $9.9 \times 10^{-3}$; (6): 0.87; (7): 0.022; (8): 1. (O) Boxplot showing the expression score of the GO term 'regulation of cholesterol biosynthetic process' within specific clusters comparing mock and 50% compressed samples, respectively. Exact $P$ values: (1): 0.41; (2): 0.87; (3): $2.1 \times 10^{-3}$; (4): 0.78; (5): 0.22; (6): 0.63; (7): 0.84; (8): 0.9. For (M, N, O), Boxplots show the median (center line), interquartile range (box), and whiskers extending to the most extreme values within 1.5× the interquartile range; dots represent individual organoids. Statistical significance was assessed using a two-sided Wilcoxon rank-sum test (ns: $P > 0.05$; *: $P < 0.05$; **: $P < 0.01$; ***: $P < 0.001$). For this analysis all neural stem and progenitor cells as well as the more differentiated neurons are included (cluster 0, 1, 3, 5, 6, 9, 11, 13 in Fig. 5A). Number of cells in each cluster (mock/50% compressed): (0): 220/1175; (1): 324/819; (3): 229/390; (5): 295/312; (6): 125/427; (9): 168/220; (11): 53/173; (13): 5/97.

