## [Peer Review File · EMBO Reports]

Mechanical impact on neural stem cell lineage decisions in human brain organoids

Sven Falk, Hanna Lampersperger, Michael Tranchina, Bastian Meth, Dandan Han, Negar Nayebzadeh, Nina Reiter, Sonja Kuth, Markus Lorke, Aldo Boccaccini, Silvia Budday, and Marisa Karow

Corresponding author(s): *Sven Falk (sven.falk@fau.de)* , *Marisa Karow (marisa.karow@fau.de)*

Review Timeline:

Transfer Date:	4th Mar 25
Editorial Decision:	10th Mar 25
Revision Received:	6th Nov 25
Editorial Decision:	11th Dec 25
Revision Received:	23rd Jan 26
Accepted:	30th Jan 26

Editor: Esther Schnapp

Transaction Report: This manuscript was transferred to EMBO reports following peer review at The EMBO Journal.

Referee #1:

This is a fantastic study by the Falk and Karow lab identifying very clear effects of mechanical forces on human neural stem cells and their basic biology. Applying controlled compression forces onto growing human cerebral organoids, the authors discover that this promotes dramatic changes on gene expression, particularly on neural stem cells, thereby changing their fate decisions toward amplification and against neurogenesis. They find that mechanical compression apparently does not affect older organoids but is specific to early organoids, which through a very clever experiment they show that these effects are very similar following acute or sustained mechanical stimulation. Then, the authors dive into the transcriptional changes caused by mechanical compression and related to neural stem cell amplification, and find thousands of differentially-expressed genes dedicated to lipid metabolism and to define telencephalic regionalization, namely. A final transcriptomic analysis at single cell level reveals no differences in cell type composition, but some differences in gene expression along developmental trajectories, importantly including again pattern specification and metabolism, as found in bulk analyses.

As the interaction between mechanics and brain development is beginning to be recognized in the field, this study will undoubtedly be a cornerstone for future research, a reference for time to come. In my opinion the study is brilliant as is, but I would recommend following some suggestions for further clarity and improvement of an already elegant piece of work.

Major comments:

The lack of apparent signs of increased cell death must be substantiated and quantified, which becomes important in the second part of the manuscript.

What does it mean that Sox2 changed most in cells closer to the apical surface? Do Sox2 levels oscillate during cell cycle, and compression affects mostly cells near M-phase?

Is the absence of response in older organoids because there is a much greater proportion of neurons, and these are less compressible? Were these compressions mechanically deforming NSCs at a scale similar to those applied to 30d organoids?

"These data highlight a so far unappreciated level of NSC regulation where mechanical stimuli influence the balance between proliferation and differentiation of NSCs." - The authors discard the alternative possibility that newborn neurons died more frequently after compression, while NSC lineage decisions remained unchanged. Refutation of this

possibility requires evidence. At first the authors mention lack of increased apoptosis, but this is not shown, nor if this would affect differently NSC vs neurons.

The manuscript would significantly benefit from including additional markers of progenitor cells in Fig 5 and EV3, including Sox2 but also Pax6, Nestin, Hes1, Top2A, etc.

Regarding single cell transcriptomic analyses, what about the thousands of DEGs identified in bulk transcriptomes and strongly related to lipid metabolism? In which clusters/cell types are they expressed? Lipid metabolism doesn't seem to outstand particularly in the scRNA-seq dataset, although it does appear there somehow. This clear inconsistency with the RNAseq results merits some explanation.

Minor points:

The grammar of the second to last sentence in Abstract needs revising.

Are the distributions in distance to apical surface and in Sox2 protein distribution significantly different? Must be indicated in Figure 1F.

Why the values of Sox2 protein in Fig 3H and I are much more variable in HG than in MG? Can the authors maybe discuss this point?

"In sum, our data provide evidence that acute mechanical manipulations alter progenitor lineage progression in developing brain organoids likely mediated through changes in the levels of the NSC factor SOX2." - Alteration of progenitor lineage progression is an overstatement here, as there is no direct evidence to support it yet.

In legend of Fig 1H, indications on p-values are confusing. What does it mean "exact P values top to down"?

Legend to Figure 1: "exact P values top to down: 1.0, 1.0, 9.399×10^{-1} ". It is not clear what this means.

Figure 5 does not show lipid metabolism, as stated in the main text, but only "cholesterol processes".

In Figure 5G, what is being measured in the y-axis?

Legend to Figure 5: there is no panel "H"; this must be corrected to "G".

Referee #3:

In their manuscript "Mechanical impact on neural stem cell lineage decisions in human brain organoids" Lampersperger and colleagues used iPSC-derived organoids and applied acute or lasting forces onto organoids at different maturation stages using a rheometer. They show that applying mechanical forces (i.e., deforming organoids) affects lineage decisions and alters molecular programs of neural cells within the organoids, as measured by bulk and single cell RNA-sequencing. The authors very nicely introduce previous work in the context of developmental biology and morphogenesis that described the clear impact of mechanical forces on a variety of biological processes. Using brain organoids to study the impact of mechanical forces is an interesting approach (that has been used in other systems in previous work, for example using intestinal organoids). The data describing altered fate decisions and transcriptional profiles are convincing.

However, it remains somewhat unclear what we learn from the study: is shape/fate of brain organoids - when they grow, using an unguided approach as applied here - dependent on different mechanical forces? Is that what happens under "physiological" conditions? Are there measurable differences in mechanical forces that affect shape/fate of progenitors or neurons (e.g., using approaches to measure forces within organoids and then to manipulate forces, for example by using different stiffness of extracellular matrix)? Are the observed changes in gene expression relevant for fate decisions? That said: the data are interesting but in its current form the study falls somewhat short to answer a biological question as the relevance of "pushing down" organoids with a rheometer and how that may "resemble" a biological process remains elusive. Thus, we feel that the data shown represent an interesting observation but that the biological relevance remains somewhat unclear.

Referee #4:

General comments

During brain morphogenesis, cells are subjected to mechanical pressure from the extracellular tissue environment. Using human brain organoids, Lampersperger et al report that mechanical stimuli by a rheometer influence gene expression in neural stem cells. The direction of this study is interesting. In particular, their data that organoid compression led

to upregulation of ventral genes along with downregulation of dorsal genes have potential implications in the field of developmental biology. However, there are concerns about the reliability of the experimental data and their significance in the physiological context.

Major concerns

1) Fig. 1: The authors conclude that acute compression of brain organoids results in SOX2 upregulation: they write, "we found a prominent increase in the overall presence of SOX2 signal across the entire sections of compressed organoids" (Fig 1B). However, the data were obtained using immunocytochemistry. Since immunocytochemistry is a qualitative and not a quantitative method, the authors should quantify the SOX2 levels in organoids by immunoblot analysis. Importantly, SOX2 mRNA did not increase in compressed cells (Fig. 5C). How do the mechanical stimuli upregulate SOX2 protein within 24h without increasing its mRNA?

2) The key point of the compression experiments is whether the magnitude of the force applied to the organoids is physiological or not. I am not sure whether neural stem cells receive large mechanical pressures from the environment that induce 40-80% mechanical compression. In this regard, the unit of the strength of the force (vertical axis) in Fig. 1D is not described. In addition, the colors of the dots in the graph do not match with those in the explanation box (also Fig. EV1D). Thus, readers cannot understand the magnitude of the force applied to the organoids.

Similarly, the authors write "During the second and third cycle of compression, the forces acting on the organoid are typically lower" (page 3, Fig. EV1B). How the authors conclude that the force described in Fig. EV1B is "low"? It should be discussed in the context of the magnitude of the force received by the cells in vivo. In addition, readers would not be able to understand what is the horizontal axis of Fig. EV1B (Stretch [-]).

In summary, since any cell expressing mechanosensitive molecules could respond to an artificially applied force, the authors need to describe the nature and magnitude of the applied force explicitly and explain its in vivo relevance.

3) The conclusion "our data illustrate that mechanical manipulations govern the transcriptional networks driving early patterning events establishing the dorso-ventral as well as the anterior-posterior axis (page 9)" requires experimental data to support it. The data that organoid compression leads to upregulation of ventral genes along with downregulation of dorsal genes is potentially interesting. However, the paper does not demonstrate the force-driven patterning of the dorso-ventral axis in the organoids. In addition, the deregulation of genes associated with anterior-posterior patterning does not support this conclusion. The authors also need to discuss their data in the context of the

magnitude of the force received by the cells during the dorso-ventral and anterior-posterior axis the patterning.

4) The units of the analyzed data should be indicated throughout the Figures and their legends (also see Major concern 2).

Minor points

1) Please discuss possible mechanosensors involved in the compression-induced gene up/down regulation.

2) Figs. 1C, E, H, 2C-F, 3A, G-I, and EV1E: What are the yellow images placed on the upper-left of the graphs? In addition, it is difficult read the letters in Fig. EV1E.

3) Page 4: F-actin, not F-ACTIN, is a standard abbreviation of actin filament.

Dear Sven,

Thank you for the transfer of your manuscript to EMBO reports and for your revision plan.

I agree with your suggestions for how to revise your ms and would thus like to invite you to revise it with the understanding that the referee concerns must be fully addressed and their suggestions taken on board. Please address all referee concerns in a complete point-by-point response. Acceptance of the manuscript will depend on a positive outcome of a second round of review. It is EMBO reports policy to allow a single round of major revision only and acceptance or rejection of the manuscript will therefore depend on the completeness of your responses included in the next, final version of the manuscript.

We realize that it is difficult to revise to a specific deadline. In the interest of protecting the conceptual advance provided by the work, we recommend a revision within 3 months (10th Jun 2025). Please discuss the revision progress ahead of this time with the editor if you require more time to complete the revisions.

- 1) A data availability section providing access to data deposited in public databases is missing. If you have not deposited any data, please add a sentence to the data availability section that explains that.
- 2) Your manuscript contains statistics and error bars based on $n=2$. Please use scatter blots in these cases. No statistics should be calculated if $n=2$.

3) We replaced Supplementary Information with Expanded View (EV) Figures and Tables that are collapsible/expandable online. A maximum of 5 EV Figures can be typeset. EV Figures should be cited as 'Figure EV1, Figure EV2' etc... in the text and their respective legends should be included in the main text after the legends of regular figures.

5) a complete author checklist, which you can download from our author guidelines <https://www.embopress.org/page/journal/14693178/authorguide>. Please insert information in the checklist that is also reflected in the manuscript. The completed author checklist will also be part of the RPF.

6) Please note that all corresponding authors are required to supply an ORCID ID for their name upon submission of a revised manuscript (<https://orcid.org/>). Please find instructions on how to link your ORCID ID to your account in our manuscript tracking system in our Author guidelines <https://www.embopress.org/page/journal/14693178/authorguide#authorshipguidelines>

12) All Materials and Methods need to be described in the main text using our 'Structured Methods' format, which is required for all research articles. According to this format, the Methods section includes a Reagents and Tools Table (listing key reagents, experimental models, software and relevant equipment and including their sources and relevant identifiers) followed by a Methods and Protocols section describing the methods using a step-by-step protocol format. The aim is to facilitate adoption of the methodologies across labs. More information on how to adhere to this format as well as a downloadable template (.docx) for the Reagents and Tools Table can be found in our author guidelines:

An example of a Method paper with Structured Methods can be found here: <https://www.embopress.org/doi/full/10.1038/s44320-024-00037-6#sec-4>

I look forward to seeing a revised form of your manuscript when it is ready.

EMBOR-2025-61463-T

Point-by-point letter for revised manuscript

We thank the reviewers for thoroughly reviewing our manuscript and providing critical insights and feedback. Based on the recommendations, we have now substantially revised the manuscript and added new data and experiments to support and strengthen our main findings. In brief, we show that the acute mechanical manipulations, do not cause increased cell death in young neurons generated after the compression and thus cell-type specific cell-death is not the reason for a reduced neuronal output in the compressed organoids. We extended the single cell RNA-seq analysis revealing cell-type specific effects of the acute mechanical manipulations. We have furthermore restructured the manuscript, revised, and edited the text accordingly.

Below we provide a summary of the major revisions and changes based on the main and supplementary figures.

Figure 1	 - replacement of previous Fig 1F with new quantifications - removal of previous Fig 1G
Figure 2	 - unchanged
Figure 3	 - addition of quantifications of cell death in compressed organoids, new Fig 3D - accordingly, all subsequent figure panels have been relabeled
Figure 4	 - unchanged
Figure 5	 - unchanged

Figure EV1	 - Fig EV1B, addition of a scheme below the graph to represent the degree of compression and replacement of the curves
Figure EV2	 - addition of example pictures of immunostainings used for assessment of apoptosis (new Fig EV2B) - additional quantification of cell death (new Fig EV2C) - Fig EV2D relabeled (previous 2B)
Figure EV3	 - addition of new plots showing the expression of additional neuron (Fig EV3C) and progenitor markers (Fig EV3D) in scRNA-seq dataset - addition of box plots showing cluster specific regulation of GO-terms (Figs EV3M-O)

Referee #1:

This is a fantastic study by the Falk and Karow lab identifying very clear effects of mechanical forces on human neural stem cells and their basic biology. Applying controlled compression forces onto growing human cerebral organoids, the authors discover that this promotes dramatic changes on gene expression, particularly on neural stem cells, thereby changing their fate decisions toward amplification and against neurogenesis. They find that mechanical compression apparently does not affect older organoids but is specific to early organoids, which through a very clever experiment they show that these effects are very similar following acute or sustained mechanical stimulation. Then, the authors dive into the transcriptional changes caused by mechanical compression and related to neural stem cell amplification and find thousands of differentially-expressed genes dedicated to lipid metabolism and to define telencephalic regionalization, namely. A final transcriptomic analysis at single cell level reveals no differences in cell type composition, but some differences in gene expression along developmental trajectories, importantly including again pattern specification and metabolism, as found in bulk analyses.

As the interaction between mechanics and brain development is beginning to be recognized in the field, this study will undoubtedly be a cornerstone for future research, a reference for time to come. In my opinion the study is brilliant as is, but I would recommend following some suggestions for further clarity and improvement of an already elegant piece of work.

We thank the reviewer for the time spent in reviewing our manuscript and for recognizing the importance of this study for the field. Moreover, we thank the reviewer for the constructive criticism which we address in detail below.

Major comments:

The lack of apparent signs of increased cell death must be substantiated and quantified, which becomes important in the second part of the manuscript.

We agree with the reviewer that cell death needs to be quantitatively assessed in our paradigms, and we therefore quantified cell death using cleaved caspase 3 (cCAS3) stainings in compressed organoids.

These new data incorporated in the manuscript (new Fig 3D, Fig EV2B, C) show that there is neither a significant difference in apoptosis in all cells born after compression (cCAS3/BrdU; Fig EV2C) nor in young neurons born after compression (cCAS3/BrdU/NEUN; Fig 3D). Hence with this new data we can now rule out that a cell type-specific increase in cell death is the underlying explanation for the changes in the cellular composition after mechanical manipulation and underscore that mechanical manipulations alter NSC lineage decisions.

What does it mean that Sox2 changed most in cells closer to the apical surface? Do Sox2 levels oscillate during cell cycle, and compression affects mostly cells near M-phase?

Yes, indeed our earlier quantification had indicated an uneven distribution of SOX2 protein levels in control organoids along the apico-basal axis in the ventricular zone like structures (VZLS) with slightly higher levels at more apical positions and lower SOX2 levels at more basal positions. In the previous version of the manuscript in Fig 1F, this was shown as density distribution which does not ideally reflect the variation of the data points, but rather emphasizes on the extreme data points. Thanks to the comment of the reviewer, we now show a new representation of the data, which allowed us to statistically address whether the reaction of the cells is different depending on the location within the VZLS. New Fig 1F shows that the upregulation of SOX2 protein levels is compression-dependent and significantly higher across the apico-basal axis of the VZLS. For this analysis we binned the VZLS in 10 equally sized bins with bin 0 at the most apical and bin 9 at the most basal site and compared the expression levels in uncompressed and compressed samples. New Fig 1F shows the data as fold change to the respective uncompressed sample. In addition, we add here (PP-Figure 1A) the absolute quantifications of the SOX2 protein levels in the same bins across conditions. As apparent from the quantifications, there are slight changes of the SOX2 levels within the conditions across bins, but they are not significant (shown in PP-Figure 1B). The levels of SOX2 in uncompressed samples are not different between apical and basal sites. Moreover, the example of 50% compression shows that the differences (SOX2 upregulation upon compression) are not different between apical and basal sites (PP-Figure 1C).

We thank the reviewer to bring up this aspect and are happy to now provide an accurate quantification of the SOX2 increase within the VZLS.

PP-Figure 1. Quantification of SOX2 protein levels within ventricular zone like structures (VZLS). **A**, Quantification of the SOX2 protein levels in 10 equally sized apico-basal bins of the VZLS across all conditions as shown by box plots and jitter. Note that irrespective of the location within the VZLS, there is a compression-dependent upregulation of SOX2. **B**, Box plots and jitter showing the levels of SOX2 protein within VZLS, binned in 10 equally sized bins from apical (bin 0) to the most basal (bin 9) site. With one exception there are no significant changes in the levels of SOX2 in uncompressed samples. **C**, Box plots and jitter showing the SOX2 protein levels as fold change to the respective uncompressed sample within the same bin along the apico-basal axis of the VZLS. Note that there are no differences across the bins which shows that the upregulation of SOX2 upon compression is not depending on the location within the VZLS.

Is the absence of response in older organoids because there is a much greater proportion of neurons, and these are less compressible? Were these compressions mechanically deforming NSCs at a scale similar to those applied to 30d organoids?

We thank the reviewer for raising this interesting aspect. The reviewer is right in that the tissue stiffness increases during neurodevelopment in general and is higher in neuronal compartments than in progenitor compartments of the developing brain (Iwashita *et al*, 2014). In our experiments we performed the same compression with the same degree of deformation at all compared developmental stages. Absence of cellular response to compression can hence not be explained by varying degrees of deformation. We will include this aspect in the discussion in more detail.

"These data highlight a so far unappreciated level of NSC regulation where mechanical stimuli influence the balance between proliferation and differentiation of NSCs." - The authors discard the alternative possibility that newborn neurons died more frequently after compression, while NSC lineage decisions remained unchanged. Refutation of this possibility requires evidence. At first the authors mention lack of increased apoptosis, but this is not shown, nor if this would affect differently NSC vs neurons.

As mentioned also above we fully agree with the reviewer that cell death needs to be assessed quantitatively in specific cell types and therefore performed immunohistochemistry against cleaved caspase 3 (cCAS3) as indicator of apoptosis, BrdU to reveal newborn cells, and NEUN to distinguish neurons from other cells.

These new data incorporated in the manuscript (new Fig 3D, Fig EV2B, C) show that there is neither a significant difference in apoptosis in all cells born after compression (cCAS3/BrdU; Fig EV2C) nor in young neurons born after compression (cCAS3/BrdU/NEUN; Fig3D). Hence with this new data we can now rule out that a cell type-specific increase in cell death is the underlying explanation for the changes in the cellular composition after mechanical manipulation and underscore that mechanical manipulations alter NSC lineage decisions.

The manuscript would significantly benefit from including additional markers of progenitor cells in Fig 5 and EV3, including Sox2 but also Pax6, Nestin, Hes1, Top2A, etc.

This is a good suggestion, and we added plots showing the expression of additional neuronal (*DCX* in Fig EV3C) and progenitor markers (*SOX2*, *NES*, *HES5*, *TOP2A* in Fig EV3D) in the scRNA-seq data.

Regarding single cell transcriptomic analyses, what about the thousands of DEGs identified in bulk transcriptomes and strongly related to lipid metabolism? In which clusters/cell types are they expressed? Lipid metabolism doesn't seem to stand out particularly in the scRNA-seq dataset, although it does appear there somehow. This clear inconsistency with the RNAseq results merits some explanation.

The reviewer is right that there are many lipid-associated GO terms based on the DE genes in the bulk RNAseq data which averages transcriptional differences over all cells present in an organoid. The big advantage of the scRNA-seq data is that we can deconvolute these differences to specific cell types. In the previous manuscript we focused the analysis of the single cell data mainly on the progenitor cells molecularly explaining the cellular effects we have observed. In Fig 5G one can appreciate that the compression induced effect on lipid metabolism (showing 'regulation of cholesterol biosynthetic processes' as one proxy) is apparent rather late in the pseudotime, i.e. in the neurons. We performed a new complementary analysis to focus on lipid-associated genes in a cell type specific manner including both progenitor and neuron clusters (cluster numbers as specified in Fig 5A). As shown in PP-Figure 2 and the new Figs EV3M-O in the revised manuscript, we analyzed patterning and metabolism associated GO terms prioritized by the bulk RNA-seq dataset (Fig 4C) and assessed the difference of the expression of the respective GO term associated genes across the clusters between mock and 50% compressed samples. These new analyses reveal that the impact of mechanical manipulation on the expression of these genes is lineage and cluster dependent. The GO terms 'pattern specification processes' and 'mitochondrial ATP synthesis coupled electron transport' are deregulated in most clusters (Fig EV3M, N). On the other hand, lipid metabolism associated GO terms show a more cluster specific deregulation. While the GO terms 'very low density lipoprotein particle clearance', 'positive regulation of lipid transport' and 'regulation of cholesterol biosynthetic process' are mainly deregulated in the neuronal clusters 3 and 6, the more general term 'cholesterol biosynthetic process' shows again a broader deregulation pattern.

Together these new analyses resolve the seeming inconsistency between the bulk- and scRNA-seq analysis and highlight the cell type-specific effects on metabolism induced by mechanical manipulations.

PP-Figure 2. Quantification of the level of expression of the genes associated with the respective GO terms. A, B, C, Box plots showing the respective GO score across clusters comparing mock and 50% compressed samples. * $P < 0.05$, ** $P < 0.01$, * $P < 0.001$. Two-sided Wilcoxon rank sum test.**

Minor points:

The grammar of the second to last sentence in Abstract needs revising.

We apologize for the mistake and adjusted the sentence accordingly.

Are the distributions in distance to apical surface and in Sox2 protein distribution significantly different? Must be indicated in Figure 1F.

We thank the reviewer for pointing this out. As indicated in the comment above with our new quantification using binning of the apico-basal axis within the VZLS, we can now say that there are no statistical differences in both the levels of SOX2 along the apico-basal axis, and the degree of upregulation upon compression (PP-Figure 1). We adapted the manuscript accordingly.

Why the values of Sox2 protein in Fig 3H and I are much more variable in HG than in MG? Can the authors maybe discuss this point?

The reviewer is right that the variations in SOX2 protein levels are higher in the HG than in the MG embedded organoids. The hydrogels that we use are stiffer than the Matrigel typically used for embedding. Presumably, the stiffer the matrix on the outside the steeper the gradient inside the organoid. This likely leads to a non-uniform response of the cells to the stiffer surrounding matrix, i.e. the closer the cells are to the outer boarder of the organoid, the more pronounced the response. If this would be the case, then this could lead to a bigger variation across samples. As the reviewer suggested, we discussed this aspect in the revised manuscript.

"In sum, our data provide evidence that acute mechanical manipulations alter progenitor lineage progression in developing brain organoids likely mediated through changes in the levels of the NSC factor SOX2." - Alteration of progenitor lineage progression is an overstatement here, as there is no direct evidence to support it yet.

We agree with the reviewer and down-toned our conclusion.

In legend of Fig 1H, indications on p-values are confusing. What does it mean "exact P values top to down"?

We apologize if this description was confusing. In fact, in this particular case none of the comparisons showed a statistically significant difference. From top to down indicates the P-values corresponding to the most upper line, the one below and then the lowest line. We clarified this in the respective figure legends.

Legend to Figure 1: "exact P values top to down: 1.0, 1.0, 9.399×10^{-1} ". It is not clear what this means.

This is the same as above. We clarified this in the figure legends.

Figure 5 does not show lipid metabolism, as stated in the main text, but only "cholesterol processes".

The reviewer is right. We adjusted this and as mentioned above added more data to lipid metabolism.

In Figure 5G, what is being measured in the y-axis?

On the y-axis we plotted the respective score of the indicated GO-term. The score reflects the average expression of a particular gene set, in this case the genes associated with the respective GO term. We changed the figure legend to render this more comprehensive.

Legend to Figure 5: there is no panel "H"; this must be corrected to "G".

We apologize for this mistake corrected the labeling accordingly.

Referee #3:

In their manuscript "Mechanical impact on neural stem cell lineage decisions in human brain organoids" Lampersperger and colleagues used iPSC-derived organoids and applied acute or lasting forces onto organoids at different maturation stages using a rheometer. They show that applying mechanical forces (i.e., deforming organoids) affects lineage decisions and alters molecular programs of neural cells within the organoids, as measured by bulk and single cell RNA-sequencing. The authors very nicely introduce previous work in the context of developmental biology and morphogenesis that described the clear impact of mechanical forces on a variety of biological processes. Using brain organoids to study the impact of mechanical forces is an interesting approach (that has been used in other systems in previous work, for example using intestinal organoids). The data describing altered fate decisions and transcriptional profiles are convincing.

First of all, we would like to thank the reviewer for taking the time to carefully read our manuscript and provide constructive comments. We are grateful that the reviewer highlights our approach as interesting and considers the -omics data convincing. We appreciate the concerns of the reviewer and answer below in a detailed point-by-point manner.

However, it remains somewhat unclear what we learn from the study: is shape/fate of brain organoids - when they grow, using an unguided approach as applied here - dependent on different mechanical forces?

We are sorry that probably there was a misunderstanding based on our previous manuscript. In our study we test how mechanical manipulations impact on the development of brain organoids. In fact, we did find that both, an acute mechanical impact as well as a long-lasting change in the mechanical environment influence the fate of brain organoid resident cells. This establishes a connection between mechanics and cell fate in a human brain tissue-like model.

We did not investigate how the (irregular) shape of organoids impacts on tissue mechanics and cell fate acquisition.

Instead, our data suggest that mechanical manipulations are sufficient to induce molecular frameworks impacting on regional identity and cellular metabolism. We adapted the manuscript to render this more clear.

Is that what happens under "physiological" conditions?

We thank the reviewer for raising this interesting aspect and we are happy to elude on this. It has been shown earlier that the tissue stiffness increases during neurodevelopment in general and is higher in neuronal compartments than in progenitor compartments of the developing

brain (Iwashita *et al*, 2014). Therefore, we are convinced that studying the consequences of mechanical impacts is an interesting and relevant avenue to pursue.

During *in utero* natural brain development of mice the intracranial pressure oscillates within a range of 150 – 1500 Pa with an oscillation period of around 33 seconds (Akaike *et al*, 2025) indicating dynamic mechanical stimuli to an extent comparable to the duration and magnitude of manipulations performed in our study. The intracranial pressure is driven by uterine contractions as well as the production of the cerebrospinal fluid (CSF) (Moazen *et al*, 2016; Jones *et al*, 1987). In chick embryos release of CSF, and by that both changing the pressure as well as the biochemical milieu, alters proliferation of progenitors in the developing CNS (Gato *et al*, 2005; Desmond & Schoenwolf, 1985). Using rheometer mediated compression, we now can disentangle the impact of mechanical aspects from biochemical factors impacting on cellular behavior. The forces applied during rheometer mediated compressions (Fig EV1B) are in the same magnitude described *in vivo*. Hence, the rheometer mediated compression allows to apply forces similar to the ones acting during brain development but for a limited time and temporally precisely controlled.

Complementary to this, the hydrogel mediated variation of the mechanical environment is providing a long-lasting change in the mechanical environment the brain organoids are developing in.

We included a section in the manuscript highlighting these physiological aspects and discuss the advantages and limitations of the approaches applied.

Are there measurable differences in mechanical forces that affect shape/fate of progenitors or neurons (e.g., using approaches to measure forces within organoids and then to manipulate forces, for example by using different stiffness of extracellular matrix)?

We agree with the reviewer that it is interesting to use varying degrees of stiffness of extracellular matrix. In fact, we are employing hydrogels with different mechanical properties, i.e. a different stiffness for organoid embedding, thereby providing a long-lasting change in the mechanical environment the organoids are growing in (Figs 3E-J). Our results show that embedding in hydrogels with different stiffnesses impacts on SOX2 neural stem cells within brain organoids.

To further illustrate that the mechanical properties of the hydrogels applied in our study, have distinct mechanical properties, we received data from our collaborators from the Boccaccini group (PP-Figure 3). The manuscript describing the mechanical and general material properties of the HGs in more detail is published (Lorke *et al*, 2025).

Figure for referee with unpublished data and its description has been removed upon request by the authors.

Are the observed changes in gene expression relevant for fate decisions?

Our data show that upon compression, there is an increase in SOX2 protein levels. Furthermore, we found functional implications to this, since SOX2 target genes were subsequently induced in compressed cells (Fig 5E). This is in line with findings showing that changes in SOX2 protein levels to a similar extent as described in our study, with higher levels promoting neural stem cell proliferation (Chew *et al*, 2005; Bylund *et al*, 2003; Graham *et al*, 2003; Kopp *et al*, 2008).

That said: the data are interesting but in its current form the study falls somewhat short to answer a biological question as the relevance of "pushing down" organoids with a rheometer and how that may "resemble" a biological process remains elusive. Thus, we feel that the data shown represent an interesting observation but that the biological relevance remains somewhat unclear.

We appreciate the clear wording of the reviewer, although we hope that with the additional data and the edits in the revised version, we can now convince the reviewer about the relevance of our data for developmental neurobiology. We do want to point out that our conclusions are not based on just the rheometer mediated large strain compressions, but also on the experiments using hydrogels with different physical properties. Embedding in these hydrogels provided a long-lasting change in the mechanical environment and also here we found changes in the number of progenitor cells and differentiating neurons further supporting our conclusions that mechanics impact on cellular fate acquisition.

As also eluded above, it is evident that a developing brain is exposed to varying intracranial pressure, i.e. mechanical stimuli. In addition, it is known that the mechanical environment changes during development (Iwashita *et al*, 2014, 2020), yet to what extent mechanics impact on the behavior of neural stem cells within a developing brain tissue has not been explored so far. In our study, we use different mechanical manipulations on brain organoids and assess on a cellular level the impact on organoid resident cells. Thereby we establish a direct connection of the mechanical manipulations and the cellular fate. Of course, such connection cannot be modelled directly in a human physiological environment *in vivo*, but the use of brain organoids represents an unprecedented suitable model system to address the role of mechanics and the behavior of neural stem cells within human developing brain tissue.

Referee #4:

General comments

During brain morphogenesis, cells are subjected to mechanical pressure from the extracellular tissue environment. Using human brain organoids, Lampersperger et al report that mechanical stimuli by a rheometer influence gene expression in neural stem cells. The direction of this study is interesting. In particular, their data that organoid compression led to upregulation of ventral genes along with downregulation of dorsal genes have potential implications in the field of developmental biology. However, there are concerns about the reliability of the experimental data and their significance in the physiological context.

We thank the reviewer for the efforts in reviewing our manuscript and for highlighting the importance of our work in the field of developmental biology. We appreciate the constructive comments and reply to them systematically below.

Major concerns

1) Fig. 1: The authors conclude that acute compression of brain organoids results in SOX2 upregulation: they write, "we found a prominent increase in the overall presence of SOX2 signal across the entire sections of compressed organoids" (Fig 1B). However, the data were obtained using immunocytochemistry. Since immunocytochemistry is a qualitative and not a quantitative method, the authors should quantify the SOX2 levels in organoids by immunoblot analysis. Importantly, SOX2 mRNA did not increase in compressed cells (Fig. 5C). How do the mechanical stimuli upregulate SOX2 protein within 24h without increasing its mRNA?

The reviewer is right that we based our conclusions on quantified immunofluorescence data. To account for the batch-to-batch variability of brain organoids, we use uncompressed organoids of the same batch as controls. All stainings of a batch are performed simultaneously in the same run and the images are acquired with the same settings, producing pictures with

pixel values only in the dynamic range of the picture, i.e. no under- or over-exposed pixel values. The quantification of the images is performed by using an automated image analysis pipeline, excluding any impact of the experimenter in the analysis. Such standardized approaches have been used to quantify protein levels in tissue before and were verified to be quantitative by mass spectrometry (Toki *et al*, 2017; Montero Llopis *et al*, 2021). Our analysis approach has the advantage that SOX2 protein levels are quantified only in SOX2 positive cells. Quantifying the SOX2 protein levels by Western Blot has the considerable disadvantage that it measures the mean SOX2 protein levels in the whole organoid. However, the fraction of SOX2 positive cells varies considerably between different organoids even of the same batch, preventing an accurate measurement of protein levels in SOX2 positive cells.

Importantly, we did detect an upregulation of transcripts of genes that are regulated by SOX2 (Fig 5E) showing not only the functional consequence of more SOX2 protein but providing proof by an independent approach for an increase in SOX2 protein levels.

In the revised manuscript we further embedded our findings in the body of literature discussing control of SOX2 protein levels on post-transcriptional levels in particular by miRNAs (Zhang *et al*, 2020).

2) The key point of the compression experiments is whether the magnitude of the force applied to the organoids is physiological or not. I am not sure whether neural stem cells receive large mechanical pressures from the environment that induce 40-80% mechanical compression. In this regard, the unit of the strength of the force (vertical axis) in Fig. 1D is not described. In addition, the colors of the dots in the graph do not match with those in the explanation box (also Fig. EV1D). Thus, readers cannot understand the magnitude of the force applied to the organoids.

During *in utero* natural brain development of mice the intracranial pressure oscillates within a range of 150 – 1500 Pa with an oscillation period of around 33 seconds (Akaike *et al*, 2025) indicating dynamic mechanical stimuli to an extent comparable to the duration and magnitude of manipulations performed in our study. The intracranial pressure is driven by uterine contractions as well as the production of the cerebrospinal fluid (CSF) (Moazen *et al*, 2016; Jones *et al*, 1987). In chick embryos release of CSF, and by that both changing the pressure as well as the biochemical milieu, alters proliferation of progenitors in the developing CNS (Gato *et al*, 2005; Desmond & Schoenwolf, 1985). Using rheometer mediated compression, we now can disentangle the impact of mechanical aspects from biochemical factors impacting on cellular behavior. The forces applied during rheometer mediated compressions (Fig EV1B) are in the same magnitude described *in vivo*. Hence, the rheometer mediated compression allows to apply forces similar to the ones acting during brain development but for a limited time and temporally precisely controlled.

Complementary to this, the hydrogel mediated variation of the mechanical environment is providing a long-lasting change in the mechanical environment the brain organoids are developing in.

We included a section in the manuscript highlighting these physiological aspects and discuss the advantages and limitations of the approaches applied.

We apologize to not have included the unit of strength of the force in Fig 1D and Fig EV1D amended the information. We also adjusted the colors in the mentioned graphs to make the results easier to grasp.

Similarly, the authors write "During the second and third cycle of compression, the forces acting on the organoid are typically lower" (page 3, Fig. EV1B). How the authors conclude that the force described in Fig. EV1B is "low"? It should be discussed in the context of the magnitude of the force received by the cells *in vivo*. In addition, readers would not be able to understand what is the horizontal axis of Fig. EV1B (Stretch [-]).

In summary, since any cell expressing mechanosensitive molecules could respond to an artificially applied force, the authors need to describe the nature and magnitude of the applied force explicitly and explain its in vivo relevance.

With our statement we did not intend to say that the forces are low in general, but they are lower as in the cycles before (black > blue > red, Fig EV1B). However, as mentioned also in the comment above, we added and discussed information on mechanical forces acting during brain development and put the rheometer experiments into perspective to allow for easier interpretation of the data.

The x-axis in Fig EV1B shows the degree of deformation (1: no deformation; 0.4: 60% compression). We added a visual explanation to the figure for easier readability.

3) The conclusion "our data illustrate that mechanical manipulations govern the transcriptional networks driving early patterning events establishing the dorso-ventral as well as the anterior-posterior axis (page 9)" requires experimental data to support it. The data that organoid compression leads to upregulation of ventral genes along with downregulation of dorsal genes is potentially interesting. However, the paper does not demonstrate the force-driven patterning of the dorso-ventral axis in the organoids. In addition, the deregulation of genes associated with anterior-posterior patterning does not support this conclusion. The authors also need to discuss their data in the context of the magnitude of the force received by the cells during the dorso-ventral and anterior-posterior axis the patterning.

The argument of the reviewer is based on an important misunderstanding requiring clarification. We apologize if this was not clear in the previous version of our study. In our manuscript we are concluding that mechanical manipulations regulate transcriptional networks that are involved in establishing the dorso-ventral as well as the anterior-posterior axis. We also state: "Importantly, compression of organoids did not result in novel clusters, or a significant alteration in the distribution to clusters as computed by scCODA (Büttner et al., 2021) but rather resulted in more subtle transcriptional changes with compressed cells clustering together with mock cells (Fig EV3A)". Hence, our data shows that mechanical manipulations do not result in more ventral or more posterior cells. The scRNA-seq data shows that while compressed cells globally retain the transcriptional fingerprints of uncompressed cells, they upregulate transcriptional networks more active in ventral and posterior parts of the developing brain. These findings are in particular interesting in light of recent publications of the Nowakowski and the Kriegstein lab (Wang *et al*, 2025; Delgado *et al*, 2022) showing that specifically in humans dorsal forebrain progenitors can produce cortical inhibitory neurons, neurons that are exclusively produced in the ventral forebrain in other species. The molecular underpinnings for the production of the 'ventral' inhibitory neurons from dorsal neural progenitor cells include differential activity of patterning processes such as the sonic hedgehog signaling pathway, similar to what we found after mechanical manipulation of neural cells. Importantly the local niches neural progenitor cells are residing in are heterogenous in their ECM composition (Pollen *et al*, 2015) varying both the local biochemical milieu as well as the local mechanical milieu (Iwashita *et al*, 2014, 2020). Our data indicate that the mechanical parameters neural progenitor cells are exposed to changes the transcriptional networks implicated in producing this cellular heterogeneity.

We discussed this aspect in more detail in the revised manuscript.

4) The units of the analyzed data should be indicated throughout the Figures and their legends (also see Major concern 2).

We apologize that the previous version was not clear and changed this accordingly in the revised manuscript.

Minor points

1) Please discuss possible mechanosensors involved in the compression-induced gene up/down regulation.

We were happy to follow this excellent suggestion and included the discussion of possible relevant mechanosensors in the revised manuscript.

2) Figs. 1C, E, H, 2C-F, 3A, G-I, and EV1E: What are the yellow images placed on the upper-left of the graphs? In addition, it is difficult to read the letters in Fig. EV1E.

The yellow images indicate whether the analysis was considering only the ventricular zone like structures and/or also neural compartments as specified in Fig EV1C. We added a description in the respective figure legends to increase clarity.

3) Page 4: F-actin, not F-ACTIN, is a standard abbreviation of actin filament.

We thank the reviewer for the suggestion and have implemented the change in the revised manuscript.

References:

- Bylund M, Andersson E, Novitsch BG & Muhr J (2003) Vertebrate neurogenesis is counteracted by Sox1–3 activity. *Nat Neurosci* 6: 1162–1168
- Chew J-L, Loh Y-H, Zhang W, Chen X, Tam W-L, Yeap L-S, Li P, Ang Y-S, Lim B, Robson P, *et al* (2005) Reciprocal Transcriptional Regulation of *Pou5f1* and *Sox2* via the Oct4/Sox2 Complex in Embryonic Stem Cells. *Mol Cell Biol* 25: 6031–6046
- Delgado RN, Allen DE, Keefe MG, Mancía Leon WR, Ziffra RS, Crouch EE, Alvarez-Buylla A & Nowakowski TJ (2022) Individual human cortical progenitors can produce excitatory and inhibitory neurons. *Nature* 601: 397–403
- Desmond ME & Schoenwolf GC (1985) Timing and positioning of occlusion of the spinal neurocele in the chick embryo. *Journal of Comparative Neurology* 235: 479–487
- Gato Á, Moro JA, Alonso MI, Bueno D, De La Mano A & Martín C (2005) Embryonic cerebrospinal fluid regulates neuroepithelial survival, proliferation, and neurogenesis in chick embryos. *Anat Rec A Discov Mol Cell Evol Biol* 284A: 475–484
- Graham V, Khudyakov J, Ellis P & Pevny L (2003) SOX2 Functions to Maintain Neural Progenitor Identity. *Neuron* 39: 749–765
- Iwashita M, Kataoka N, Toida K & Kosodo Y (2014) Systematic profiling of spatiotemporal tissue and cellular stiffness in the developing brain. *Development* 141: 3793–3798
- Iwashita M, Nomura T, Suetsugu T, Matsuzaki F, Kojima S & Kosodo Y (2020) Comparative Analysis of Brain Stiffness Among Amniotes Using Glyoxal Fixation and Atomic Force Microscopy. *Front Cell Dev Biol* 8: 574619
- Jones HC, Deane R & Bucknall RM (1987) Developmental changes in cerebrospinal fluid pressure and resistance to absorption in rats. *Developmental Brain Research* 33: 23–30
- Kopp JL, Ormsbee BD, Desler M & Rizzino A (2008) Small Increases in the Level of Sox2 Trigger the Differentiation of Mouse Embryonic Stem Cells. *Stem Cells* 26: 903–911
- Moazen M, Alazmani A, Rafferty K, Liu Z-J, Gustafson J, Cunningham ML, Fagan MJ & Herring SW (2016) Intracranial pressure changes during mouse development. *J Biomech* 49: 123–126
- Montero Llopis P, Senft RA, Ross-Elliott TJ, Stephansky R, Keeley DP, Koshar P, Marqués G, Gao Y-S, Carlson BR, Pengo T, *et al* (2021) Best practices and tools for reporting reproducible fluorescence microscopy methods. *Nat Methods* 18: 1463–1476
- Pollen AA, Nowakowski TJ, Chen J, Retallack H, Sandoval-Espinosa C, Nicholas CR, Shuga J, Liu SJ, Oldham MC, Diaz A, *et al* (2015) Molecular Identity of Human Outer Radial Glia during Cortical Development. *Cell* 163: 55–67
- Toki MI, Cecchi F, Hembrough T, Syrigos KN & Rimm DL (2017) Proof of the quantitative potential of immunofluorescence by mass spectrometry. *Laboratory Investigation* 97: 329–334

Wang L, Wang C, Moriano JA, Chen S, Zuo G, Cebrián-Silla A, Zhang S, Mukhtar T, Wang S, Song M, *et al* (2025) Molecular and cellular dynamics of the developing human neocortex. *Nature*

Zhang S, Xiong X & Sun Y (2020) Functional characterization of SOX2 as an anticancer target. *Signal Transduct Target Ther* 5: 135

Dear Dr. Falk,

We have now received the enclosed referee reports, as well as cross-comments from referee 3, and I am happy to say that we can offer to publish your manuscript after some more minor revisions. Please address all referee comments in the revised ms along the lines suggested by referee 3 and also please co-submit a point-by-point response to all final comments with your final ms. It is important that all final comments are addressed in the ms text, please let me know if you have any questions.

A few editorial requests will also need to be addressed:

- Please add up to 5 keywords to your ms file.
- The Code Availability and Correspondence and requests for materials section should be removed from the Data Availability Section (DAS), and the DAS needs to be moved to before the Acknowledgments.
- Affiliation #5 might be a company; employment in a biotech company should be stated in the DCIS
- The author credits need to be removed from the ms file. All credits are entered during online ms submission.

* Figure Legends - Comments *

- Please note that the exact p values are not provided in the legends of figures 1F, EV3 M-O, please provide exact p-values as reasonable.
- Please indicate the statistical test used for data analysis in the legend of figure 5E
- Please note that information related to n is missing in the legends of figures 5C, EV3 M-O

EMBO press papers are accompanied online by A) a short (1-2 sentences) summary of the findings and their significance, B) 2-3 bullet points highlighting key results and C) a synopsis image that is exactly 550 pixels wide and 200-600 pixels high (the height is variable). The synopsis image should provide a sketch of the major findings, like a graphical abstract. Please note that text needs to be readable at the final size. Please send us this information along with the final manuscript.

Referee #1:

In their point-by-point response the authors editorially addressed our previous concerns. The revised study benefits from an extended discussion/intro to the field of mechanical forces in relation to brain development and fate decisions. We still believe that studying 'pressure-dependent' gene expression and its functional relevance will substantially strengthen the manuscript (also in light of our remaining 'doubts' of how biologically relevant applied forces are in the context of in vivo brain development). That said: we do not question that this is an interesting study that will be of some value to the field.

Referee #2:

The authors partially addressed my comments in this version, but some important concerns remain unaddressed. Please see below for my original concerns, the authors' responses, and my comments on their responses.

Original concern 1) Fig. 1: The authors conclude that acute compression of brain organoids results in SOX2 upregulation: they write, "we found a prominent increase in the overall presence of SOX2 signal across the entire sections of compressed organoids" (Fig 1B). However, the data were obtained using immunocytochemistry. Since immunocytochemistry is a qualitative and not a quantitative method, the authors should quantify the SOX2 levels in organoids by immunoblot analysis. Importantly, SOX2 mRNA did not increase in compressed cells (Fig. 5C).

Authors' answer 1-1) ... All stainings of a batch are performed simultaneously in the same run and the images are acquired with the same settings, producing pictures with pixel values only in the dynamic range of the picture, i.e. no under- or over-exposed pixel values. The quantification of the images is performed by using an automated image analysis pipeline, excluding any impact of the experimenter in the analysis. Such standardized approaches have been used to quantify protein levels in tissue before and were verified to be quantitative by mass spectrometry (Toki et al, 2017; Montero Llopis et al, 2021).

Comment 1-1) I do not understand why the authors skip the standard and reliable protein quantification analysis. Immunoblot analysis is not difficult and also demonstrate the specificity of the antibody (namely, single band detected in the SDS-PAGE sample).

Authors' answer 1-2) Our analysis approach has the advantage that SOX2 protein levels are quantified only in SOX2 positive cells. Quantifying the SOX2 protein levels by Western Blot has the considerable disadvantage that it measures the mean SOX2 protein levels in the whole organoid. However, the fraction of SOX2 positive cells varies considerably between different organoids even of the same batch, preventing an accurate measurement of protein levels in SOX2 positive cells.

Comment 1-2) The measurement of SOX2 protein levels in the whole organoid provides reliable and important information. Since the authors wrote that "we found a prominent increase in the overall presence of SOX2 signal across the entire sections of compressed organoids", I think such a prominent increase could be detected by immunoblot. If the fraction of SOX2-positive cells varies considerably between different organoids, as the authors mention, it is still not convincing that the mechanical stimuli elevate SOX2 protein levels. Notably, SOX2 mRNA did not increase in compressed cells. Given that the increase in SOX2 protein levels by mechanical stimuli is the core data in this paper, I recommend obtaining compelling evidence by immunoblot analysis.

Original concern 2) How do the mechanical stimuli upregulate SOX2 protein within 24h without increasing its mRNA?

Authors' answer 2) In the revised manuscript we further embedded our findings in the body of literature discussing control of SOX2 protein levels on post-transcriptional levels in particular by miRNAs (Zhang et al, 2020).

Comment 2) In the revised version, I found that the paper Zhang et al (2020) is cited, but do not find the authors' explanation or discussion of how the mechanical stimuli could upregulate SOX2 protein without increasing its mRNA.

Original comment 3) The units of the analyzed data should be indicated throughout the Figures and their legends (also see Major concern 2).

Authors' answer 3) We apologize that the previous version was not clear and changed this accordingly in the revised manuscript.

Comment 3) The units are still lacking in Figs. 1C, E-G, 2C-F, 3A, C, H-J, Figs. EV1E, 2C, and D. If the vertical axes represent relative values, this should be explained.

Referee #3:

The authors have done a great effort at addressing my previous comments, for which I congratulate them. However, some of my previous concerns have not been completely or properly addressed, as I indicate next:

"The lack of apparent signs of increased cell death must be substantiated and quantified, which becomes important in the second part of the manuscript."

The results of analysis of caspase seem quite different between compressed and mock, although the authors indicate a statistics with a surprising $p=0,34$. Which statistical test was used?

"Regarding single cell transcriptomic analyses, what about the thousands of DEGs identified in bulk transcriptomes and strongly related to lipid metabolism? In which clusters/cell types are they expressed? Lipid metabolism doesn't seem to outstand particularly in the scRNA-seq dataset, although it does appear there somehow. This clear inconsistency with the RNAseq results merits some explanation."

Nice additional analyses have been done to address my point, but the inconsistency remains and is confirmed. Lipid metabolism gene expression changes are either broad or neuron-biased, while the compression effects were largest in young than in old organoids, presumably containing more neural Stem cells than neurons...

"Legend to Figure 1: "exact P values top to down: 1.0, 1.0, 9.399x10⁻¹". It is not clear what this means."

This has been slightly modified but continues to be completely unclear. I don't understand what the authors refer to when indicating "Most upper line, the one below and then the lowest line". What lines? Highest and lowest what? Top and bottom of what?

What the legend now states does not clarify anything: "exact P values of the comparisons as indicated in the graph. Highest, middle, lowest (top to down): 1.0, 1.0, 9.399x10⁻¹."

"In Figure 5G, what is being measured in the y-axis?"

Now that the legend explains well what the y-axis of these plots represents, the y-axis title of these plots should indicate the variable being measured (gene expression score), rather than repeating the GO term being analyzed as shown now, which is a repetition of what is already indicated in the plot title.

Cross-comments from referee 3:

In my opinion, the authors final response to reviewer 2 concern on analysis of Sox2 levels is satisfactory, especially well responded in the second round:

"Authors' answer 1-2) Our analysis approach has the advantage that SOX2 protein levels are quantified only in SOX2 positive cells. Quantifying the SOX2 protein levels by Western Blot has the considerable disadvantage that it measures the mean SOX2 protein levels in the whole organoid. However, the fraction of SOX2 positive cells varies considerably between different organoids even of the same batch, preventing an accurate measurement of protein levels in SOX2 positive cells."

While this is far from ideal, and the reviewer has a point in that levels of immunostain may differ considerably between samples, the consistent replication of the same result across a number of cases is the decisive factor here. Doing what the reviewer suggests would not be any better (likely worse), for the reasons exposed by the authors.

I do agree with reviewer 2 that the authors should add some discussion on how Sox2 protein levels may vary in the absence of changes in mRNA, beyond solely citing Zhang et al 2020, as it seems to be the case now.

EMBOR-2025-61463V2

Point-by-point letter for revised manuscript

We thank the reviewers for thoroughly reviewing our manuscript and providing critical insights and feedback. Based on the recommendations, we have revised the manuscript and changed some of the figures to better represent our findings, adapted the main text and the figure legends. The major changes are summarized here in brief:

- We added a paragraph discussing how SOX2 protein levels can be regulated on a posttranscriptional level via mechanosensitive miRNAs.
- We adapted the main text to resolve the seeming inconsistency between the bulk and single cell RNAseq data.
- We changed the way how we link the exact p-values in the figure legends to the corresponding comparison in the plots of the figures.
- We adapted the figure legends to explicitly and unambiguously state which relative values are shown in the figures.

Editorial decision letter:

We have now received the enclosed referee reports, as well as cross-comments from referee 3, and I am happy to say that we can offer to publish your manuscript after some more minor revisions. Please address all referee comments in the revised ms along the lines suggested by referee 3 and also please co-submit a point-by-point response to all final comments with your final ms. It is important that all final comments are addressed in the ms text, please let me know if you have any questions.

A few editorial requests will also need to be addressed:

- Please add up to 5 keywords to your ms file.

We added the 5 keywords: mechanics; neural stem cells; lineage decisions; metabolism; brain organoids, after the author affiliations and before the abstract to the MS.

- The Code Availability and Correspondence and requests for materials section should be removed from the Data Availability Section (DAS), and the DAS needs to be moved to before the Acknowledgments.

We removed the *code availability* and *Correspondence and requests for materials* section from the *Data availability* section. As requested, we moved the DAS to before the Acknowledgments in the MS.

- Affiliation #5 might be a company; employment in a biotech company should be stated in the DCIS

One of the authors began employment at a biotech company after completing the work for this manuscript. While she was working on the project, she was only employed at FAU Erlangen - Nürnberg. To reflect this and to avoid confusion, affiliation #5 has been removed.

- The author credits need to be removed from the ms file. All credits are entered during online ms submission.

We removed the *Author contribution* section from the MS.

* Figure Legends - Comments *

- Please note that the exact p values are not provided in the legends of figures 1F, EV3 M-O, please provide exact p-values as reasonable.

We added the exact p-values in the figure legends of Figure 1F, 5E and EV3 M-O

- Please indicate the statistical test used for data analysis in the legend of figure 5E

We added the information on the statistical test used in Figure 5E.

- Please note that information related to n is missing in the legends of figures 5C, EV3

M-O

We added the information the number of n in Figure 5C and EV3M-O.

EMBO press papers are accompanied online by

A) a short (1-2 sentences) summary of the findings and their significance

In the developing human brain mechanical cues vary across regions and along differentiation trajectories. In this study we show that acute mechanical manipulations as well as persistent changes in the mechanical environment instruct neural stem cell lineage decisions and thereby orchestrate brain development.

B) 2-3 bullet points highlighting key results and

- mechanical manipulations of brain organoids shifts the balance between proliferation and differentiation of neural stem cells.
- protein levels of the neural stem cell controlling factor SOX2 are regulated by mechanical manipulation
- mechanical manipulation impacts on molecular programs governing early patterning events and cellular metabolism

C) a synopsis image that is exactly 550 pixels wide and 200-600 pixels high (the height is variable). The synopsis image should provide a sketch of the major findings, like a graphical abstract. Please note that text needs to be readable at the final size. Please send us this information along with the final manuscript.

We provide now a synopsis image:

Referee

#1:

In their point-by-point response the authors editorially addressed our previous concerns. The revised study benefits from an extended discussion/intro to the field of mechanical forces in relation to brain development and fate decisions. We still believe that studying 'pressure-dependent' gene expression and its functional relevance will substantially strengthen the manuscript (also in light of our remaining 'doubts' of how biologically relevant applied forces are in the context of in vivo brain development). That said: we do not question that this is an interesting study that will be of some value to the field.

We thank the reviewer for their careful evaluation of our manuscript and for recognizing both the improvements in our revision and the relevance of our study to the field.

Referee

#2:

The authors partially addressed my comments in this version, but some important concerns remain unaddressed. Please see below for my original concerns, the authors' responses, and my comments on their responses.

We thank the reviewer for the effort to carefully evaluate our manuscript and for recognizing the improvements in our revision.

Original concern 1) Fig. 1: The authors conclude that acute compression of brain organoids results in SOX2 upregulation: they write, "we found a prominent increase in the overall presence of SOX2 signal across the entire sections of compressed organoids" (Fig 1B). However, the data were obtained using immunocytochemistry. Since immunocytochemistry is a qualitative and not a quantitative method, the authors should quantify the SOX2 levels in organoids by immunoblot analysis. Importantly, SOX2 mRNA did not increase in compressed cells (Fig. 5C)

Authors' answer 1-1) ... All stainings of a batch are performed simultaneously in the same run and the images are acquired with the same settings, producing pictures with pixel values only in the dynamic range of the picture, i.e. no under- or over-exposed pixel values. The quantification of the images is performed by using an automated image analysis pipeline, excluding any impact of the experimenter in the analysis. Such standardized approaches have been used to quantify protein levels in tissue before and were verified to be quantitative by mass spectrometry (Toki et al, 2017; Montero Llopis et al, 2021).

Comment 1-1) I do not understand why the authors skip the standard and reliable protein quantification analysis. Immunoblot analysis is not difficult and also demonstrate the specificity of the antibody (namely, single band detected in the SDS-PAGE sample).

Authors' answer 1-2) Our analysis approach has the advantage that SOX2 protein levels are quantified only in SOX2 positive cells. Quantifying the SOX2 protein levels by Western Blot has the considerable disadvantage that it measures the mean SOX2 protein levels in the whole organoid. However, the fraction of SOX2 positive cells varies considerably between different organoids even of the same batch, preventing an accurate measurement of protein levels in SOX2 positive cells. Comment 1-2) The measurement of SOX2 protein levels in the whole organoid provides reliable and important information. Since the authors wrote that "we found a prominent increase in the overall presence of SOX2 signal across the entire sections of compressed organoids", I think such a prominent increase could be detected by immunoblot. If the fraction of SOX2-positive cells varies considerably between different organoids, as the authors mention, it is still not convincing that the mechanical stimuli elevate SOX2 protein levels. Notably, SOX2 mRNA did not increase in compressed cells. Given that the increase in SOX2 protein levels by mechanical stimuli is the core data in this paper, I recommend obtaining compelling evidence by immunoblot analysis.

We agree with reviewer 3 that the consistent replication of the same result across multiple samples is essential for our analysis. The big advantage of our approach is that we quantify protein levels only in SOX2 positive cells and do not average across whole organoids. This is particularly important because the number of SOX2 positive cells varies between organoids and organoid batches.

Methods such as immunoblot analysis, while they have clear advantages as stated by the reviewer, would average the SOX2 protein levels across whole organoid and not only measure SOX2 protein levels in SOX2 positive cells. The variation of SOX2 positive cells from organoid to organoid combined with the variation in the response to

the mechanical stimuli and the extent of the responds renders such approaches difficult to interpret.

Importantly the single cell RNAseq analysis shows with a completely independent method that SOX2 target genes are up regulated upon compression in SOX2 positive cells corroborates these findings.

Original concern 2) How do the mechanical stimuli upregulate SOX2 protein within 24h without increasing its mRNA?

Authors' answer 2) In the revised manuscript we further embedded our findings in the body of literature discussing control of SOX2 protein levels on post-transcriptional levels in particular by miRNAs (Zhang et al, 2020).

Comment 2) In the revised version, I found that the paper Zhang et al (2020) is cited, but do not find the authors' explanation or discussion of how the mechanical stimuli could upregulate SOX2 protein without increasing its mRNA.

We agree with reviewer 2 and 3 that the discussion on how SOX2 protein levels can be regulated through miRNAs was rather short.

We now considerably expanded this aspect in the revised version, linking our results to the discovery of mechanosensitive miRNAs and discussing possible mechanotransduction pathways how mechanical stimuli might regulate miRNAs that target SOX2 mRNA.

Original comment 3) The units of the analyzed data should be indicated throughout the Figures and their legends (also see Major concern 2).

Authors' answer 3) We apologize that the previous version was not clear and changed this accordingly in the revised manuscript.

Comment 3) The units are still lacking in Figs. 1C, E-G, 2C-F, 3A, C, H-J, Figs. EV1E, 2C, and D. If the vertical axes represent relative values, this should be explained.

We agree with the reviewer that this information was not readily accessible in the figures or figure legends but rather hidden in the methods section. But even with this information in some plots the y-axis remained ambiguous. We now corrected this and changed the figure legends to explicitly state how the relative values shown were calculated.

Referee

#3:

The authors have done a great effort at addressing my previous comments, for which I congratulate them. However, some of my previous concerns have not been completely or properly addressed, as I indicate next:

We thank the reviewer for the time and effort in evaluating our manuscript, and for acknowledging the improvements made in the revised version.

"The lack of apparent signs of increased cell death must be substantiated and quantified, which becomes important in the second part of the manuscript."

The results of analysis of caspase seem quite different between compressed and mock, although the authors indicate a statistics with a surprising $p=0,34$. Which statistical test was used?

We used the non-parametric two-sided Wilcoxon rank sum test for the statistical analysis of the cleaved caspase 3 analysis in Fig 3D (and for most of the other analysis throughout the paper). Indeed, the mock group has one measurement point that is somewhat further away. However, even when excluding this datapoint from the analysis the results are not significant with a p-value of 0.11.

Alternative statistical tests:

- The results of the parametric students t-test (without showing that the data is normally distributed) is $p=0.5$.
- One-way ANOVA with Tukey's post-hoc test results in $p=0.5$.

"Regarding single cell transcriptomic analyses, what about the thousands of DEGs identified in bulk transcriptomes and strongly related to lipid metabolism? In which clusters/cell types are they expressed? Lipid metabolism doesn't seem to stand out particularly in the scRNA-seq dataset, although it does appear there somehow. This clear inconsistency with the RNAseq results merits some explanation." Nice additional analyses have been done to address my point, but the inconsistency remains and is confirmed. Lipid metabolism gene expression changes are either broad or neuron-biased, while the compression effects were largest in young than in old organoids, presumably containing more neural Stem cells than neurons...

Using the scRNA-seq data we deconvolute the transcriptional differences observed in the bulk RNAseq data to specific cell types. In Figs EV3M-O in the revised manuscript, we analyzed patterning and metabolism associated GO terms prioritized by the bulk RNA-seq dataset (Fig 4C) and assessed the difference of the expression of the respective GO term associated genes across the clusters between mock and 50% compressed samples. These analyses revealed that the impact of mechanical manipulation on the expression of these genes is lineage and cluster dependent. The GO terms 'pattern specification processes' and 'mitochondrial ATP synthesis coupled electron transport' are deregulated in most clusters (Fig EV3M, N). On the other hand, lipid metabolism associated GO terms show a more cluster specific deregulation. While the GO terms 'very low density lipoprotein particle clearance', 'positive regulation of lipid transport' and 'regulation of cholesterol biosynthetic process' are mainly deregulated in the neuronal clusters 3 and 6, the more general term 'cholesterol biosynthetic process' shows again a broader deregulation pattern (Fig EV3M-O and pp_Fig1).

These analyses reveal in which cells type the changes in lipid metabolism associated genes observed in the bulkRNAseq mainly occurs, resolving the seeming inconsistency between bulk and scRNAseq.

We absolutely agree with the reviewer that these analyses show the lipid metabolism associated changes are rather neuron-biased. We also agree with the reviewer that the main cellular effects we observe occur in neural stem and progenitor cells. We don't want to imply that the changes in lipid-metabolism are directly responsible for the changes of cellular behavior in neural stem and progenitor cells. To avoid this confusion, we adapted the main text to explicitly state this.

PP-Figure 1. Quantification of the level of expression of the genes associated with the respective GO terms. A, B, C, Box plots showing the respective GO score across clusters comparing mock and 50% compressed samples. * $P < 0.05$, ** $P < 0.01$, * $P < 0.001$. Two-sided Wilcoxon rank sum test.**

"Legend to Figure 1: "exact P values top to down: 1.0, 1.0, 9.399×10^{-1} ". It is not clear what this means."

This has been slightly modified but continues to be completely unclear. I don't understand what the authors refer to when indicating "Most upper line, the one below and then the lowest line". What lines? Highest and lowest what? Top and bottom of what?

What the legend now states does not clarify anything: "exact P values of the comparisons as indicated in the graph. Highest, middle, lowest (top to down): 1.0, 1.0, 9.399×10^{-1} ."

We apologize that the indication of the p-values in the figure legends remained unclear in the last revision (R1). In this new revised version R2 we adopted a completely different way how to link the statistical comparison in the plots with the indication of the exact p-values in the figure legends. We now explicitly number the statistical comparisons in each plot next to the indication of significance (e.g. ***(1)) and call this again in the figure legend (e.g. exact p-values: (1): 2.3×10^{-6}) to avoid any confusion.

Cross-comments from referee 3:

In my opinion, the authors final response to reviewer 2 concern on analysis of Sox2 levels is satisfactory, especially well responded in the second round: "Authors' answer 1-2) Our analysis approach has the advantage that SOX2 protein levels are quantified only in SOX2 positive cells. Quantifying the SOX2 protein levels by Western Blot has the considerable disadvantage that it measures the mean SOX2 protein levels in the whole organoid. However, the fraction of SOX2 positive cells varies considerably between different organoids even of the same batch, preventing an accurate measurement of protein levels in SOX2 positive cells." While this is far from ideal, and the reviewer has a point in that levels of immunostain may differ considerably between samples, the consistent replication of the same result across a number of cases is the decisive factor here. Doing what the reviewer suggests would not be any better (likely worse), for the reasons exposed by the authors.

We agree with reviewer 3 that the consistent replication of the same result across multiple samples is essential for our analysis. The big advantage of our approach is that we quantify protein levels only in SOX2 positive cells and do not average across whole organoids. This is particularly important because the number of SOX2 positive cells varies between organoids and organoid batches.

Importantly the single cell RNAseq analysis shows with a completely independent method that SOX2 target genes are up regulated upon compression in SOX2 positive cells corroborates these findings.

I do agree with reviewer 2 that the authors should add some discussion on how Sox2 protein levels may vary in the absence of changes in mRNA, beyond solely citing Zhang et al 2020, as it seems to be the case now.

We agree with reviewer 2 and 3 that the discussion on how SOX2 protein levels can be regulated through miRNAs was rather short.

We now considerably expanded this aspect in the revised version, linking our results to the discovery of mechanosensitive miRNAs and discussing possible mechanotransduction pathways how mechanical stimuli might regulate miRNAs that target SOX2 mRNA.

Dr. Sven Falk
University of Erlangen-Nuremberg
Institute of Biochemistry
Fahrstrasse 17
Erlangen, Bavaria 91054
Germany

Dear Sven,

I am very pleased to accept your manuscript for publication in the next available issue of EMBO reports. Thank you for your contribution to our journal.

You may qualify for financial assistance for your publication charges - either via a Springer Nature fully open access agreement or an EMBO initiative. Check your eligibility: <https://link.springer.com/journal/44319/how-to-publish-with-us>

>>> Please note that it is EMBO Reports policy for the transcript of the editorial process (containing referee reports and your response letter) to be published as an online supplement to each paper. If you do NOT want this, you will need to inform the Editorial Office via email immediately. More information is available here: <https://link.springer.com/partners/embo-press/editorial-policies#Peer%20review>